# Strong homeostatic TCR signals induce formation of self-tolerant virtual memory CD8 T cells

Ales Drobek[1,†] iD, Alena Moudra[1,†], Daniel Mueller[2], Martina Huranova[1] iD, Veronika Horkova[1], Michaela Pribikova[1], Robert Ivanek[2,3], Susanne Oberle[4,‡], Dietmar Zehn[4,5], Kathy D McCoy[6,§], Peter Draber[1] & Ondrej Stepanek[1,2,*] iD

## Abstract

Virtual memory T cells are foreign antigen-inexperienced T cells that have acquired memory-like phenotype and constitute 10–20% of all peripheral CD8[+] T cells in mice. Their origin, biological roles, and relationship to naïve and foreign antigen-experienced memory T cells are incompletely understood. By analyzing T-cell receptor repertoires and using retrogenic monoclonal T-cell populations, we demonstrate that the virtual memory T-cell formation is a so far unappreciated cell fate decision checkpoint. We describe two molecular mechanisms driving the formation of virtual memory T cells. First, virtual memory T cells originate exclusively from strongly self-reactive T cells. Second, the stoichiometry of the CD8 interaction with Lck regulates the size of the virtual memory T-cell compartment via modulating the self-reactivity of individual T cells. Although virtual memory T cells descend from the highly self-reactive clones and acquire a partial memory program, they are not more potent in inducing experimental autoimmune diabetes than naïve T cells. These data underline the importance of the variable level of self-reactivity in polyclonal T cells for the generation of functional T-cell diversity.

Keywords gene expression profiling of T-cell subsets; retrogenic T cell; self-reactivity; T-cell receptor repertoire; virtual memory T cells
Subject Categories Immunology
The EMBO Journal (2018) 37: e98518

See also: EN Truckenbrod & SC Jameson (July 2018)

## Introduction

Immunological memory is one of the hallmarks of adaptive immunity. During infection, pathogen-specific naïve T cells differentiate into short-lived effector and memory T cells. The latter facilitate long-standing protection against a secondary infection of the same pathogen. CD8[+] CD44[+] CD62L[+] central memory (CM) T cells have the capability of expansion, self-renewal, and generation of cytotoxic effector T cells upon repeated encountering of their cognate antigen (Graef et al, 2014).

Interestingly, some T cells with an apparent memory phenotype are specific to antigens which the host organism has not been exposed to (Haluszczak et al, 2009; Su et al, 2013). There are two possible explanations of their origin: (i) They could be cross-reactive T cells that have encountered another foreign cognate antigen previously (Su et al, 2013) or (ii) they are generated via homeostatic mechanisms independently of the exposure to any foreign antigens (Haluszczak et al, 2009). The strong evidence for the role of homeostatic mechanisms in generation of CD8[+] memory-like T cells was provided by the detection of these cells in germ-free mice. Since these T cells had limited prior exposure to foreign antigens, they were called virtual memory (VM) T cells (Haluszczak et al, 2009).

Generation and/or maintenance of VM T cells depends on transcription factors Eomes and IRF4, type I interferon signaling, IL-15 and/or IL-4 signaling, and CD8α[+] dendritic cells (Akue et al, 2012; Sosinowski et al, 2013; Kurzweil et al, 2014; Tripathi et al, 2016; White et al, 2016). It has been shown that VM T cells express slightly higher levels of CD122 (IL-2Rβ) and lower levels of CD49d (integrin α4, a subunit of VLA-4) than true (i.e., foreign antigen-experienced) CM T cells (Sosinowski et al, 2013). Based on these markers, VM T cells constitute for a majority of memory-phenotype CD8[+] T cells in unimmunized mice and around 10–20% of total CD8[+] T cells in lymphoid organs. Moreover, it has been proposed that memory-phenotype T cells, that accumulate in aged mice, are VM T cells (Chiu et al, 2013). However, with

1   Laboratory of Adaptive Immunity, Institute of Molecular Genetics of the Czech Academy of Sciences, Prague, Czech Republic
2   Department of Biomedicine, University Hospital and University of Basel, Basel, Switzerland
3   Swiss Institute of Bioinformatics, Basel, Switzerland
4   Swiss Vaccine Research Institute, Epalinges, Switzerland
5   Division of Animal Physiology and Immunology, School of Life Sciences Weihenstephan, Technical University of Munich, Freising, Germany
6   Department of Clinical Research (DKF), Inselspital, University of Bern, Bern, Switzerland
    *Corresponding author. Tel.: +420 241062155; E-mail: ondrej.stepanek@img.cas.cz
    †These authors contributed equally to this work
    ‡Present address: Sanofi Genzyme, Baar, Switzerland
    §Present address: Department of Physiology and Pharmacology, Cumming School of Medicine, University of Calgary, Calgary, AB, Canada

the notable exception of the very initial study that identified CD44$^+$ CD62L$^+$ CM T cells in germ-free mice (Haluszczak *et al*, 2009), all other published experiments used specific pathogen-free (SPF) mice that have significant exposure to microbial antigens. There are three major hypothesis of how virtual memory T cells might be formed in a homeostatic manner: (i) The differentiation into VM T cells occurs purely on a stochastic basis, (ii) lymphopenic environment in newborns induces differentiation of the first wave of thymic emigrants into VM T cells (Akue *et al*, 2012), or (iii) the VM T cells are generated from relatively highly self-reactive T cells that receive strong homeostatic TCR signals at the periphery (White *et al*, 2016). However, none of these hypotheses has been addressed in detail.

Although VM T cells form a large CD8$^+$ T cell population, their biological role is still unknown. VM T cells share some functional properties with true CM cells, including rapid production of IFNγ upon stimulation with a cognate antigen or cytokines (Haluszczak *et al*, 2009; Lee *et al*, 2013). On a per cell basis, ovalbumin-specific VM T cells provide a protection to ovalbumin-expressing *Listeria monocytogenes* (Lm), comparable to true CM T cells, and surpass naïve T cells with the same specificity (Lee *et al*, 2013). In addition, it has been proposed that VM T cells are capable of by-stander protection against infection, i.e., independently of their cognate antigen exposure (Chu *et al*, 2013; White *et al*, 2016). Altogether, these data pointed toward the superior role of VM T cells in protective immunity to infection. In a marked contrast to the above-mentioned findings, Decman *et al* showed that CD44$^+$ CD8$^+$ T-cell receptor (TCR) transgenic T cells isolated from unprimed mice (i.e., putative VM T cells) expand less than CD44$^-$ CD8$^+$ T cells expressing the same TCR upon antigenic stimulation *in vivo* (Decman *et al*, 2012). Likewise, VM T cells from aged mice were shown to be hyporesponsive to their cognate antigens in comparison with their naïve counterparts, mostly because of their susceptibility to apoptosis (Decman *et al*, 2012; Renkema *et al*, 2014).

As VM T cells develop independently of infection, the understanding of mechanisms that guide their development is critical in order to elucidate their biological roles. One hint is the observation that levels of CD5 (a marker of self-reactivity) on naïve T cells correlate with their ability to differentiate into VM T cells (White *et al*, 2016). In this study, we demonstrate that virtual memory T cells originate exclusively from relatively highly self-reactive T-cell clones and acquire only a partial memory gene expression program. Moreover, the interaction between CD8 and Lck (and possibly the overall intrinsic sensitivity of the TCR signaling machinery) determines the size of the virtual memory compartment. These data highlight the virtual memory T-cell formation as a T-cell fate decision checkpoint, when the intensity of TCR signals induced by self-antigens plays a central role in the decision-making process. Although virtual memory T cells show augmented responses to their foreign cognate antigen in some experimental setups in comparison with naïve T cells, the potency of VM T cells to induce pathology in an experimental model of autoimmune diabetes is not higher than that of naïve T cells.

## Results

### Strong homeostatic TCR signaling induces virtual memory T cells

In this work, we aimed to understand why some mature CD8$^+$ T cells differentiate into VM T cells and some maintain their naïve

phenotype. The peripheral T-cell pool of polyclonal mice consists of clones with different level of self-reactivity. To test the hypothesis that the level of self-reactivity plays a role in the formation of virtual memory T cells, we took advantage of previous observations that coupling frequency (or stoichiometry) of CD8 coreceptor to Lck kinase is a limiting factor for the TCR signaling in thymocytes (Erman *et al*, 2006; Stepanek *et al*, 2014). We used T cells from CD8.4 knock-in mouse strain, that express a chimeric CD8.4 coreceptor, consisting of the extracellular portion of CD8α fused to the intracellular part of CD4 (Erman *et al*, 2006). In comparison with WT CD8α, the chimeric CD8.4 coreceptor is more strongly binding Lck, a kinase initiating the TCR signal transduction. As a consequence, higher fraction of CD8.4 coreceptors molecules than CD8 coreceptors are loaded with Lck (Erman *et al*, 2006; Stepanek *et al*, 2014). Because the self-antigen triggered TCR signaling is stronger in CD8.4 T cells than CD8WT T cells (Park *et al*, 2007; Kimura *et al*, 2013; Stepanek *et al*, 2014), we use the CD8.4 T cells as a model to address the role of self-reactivity in virtual memory T-cell formation.

First, we compared monoclonal F5 Rag$^{-/-}$ T cells (henceforth CD8WT F5) and CD8.4 knock-in F5 Rag$^{-/-}$ T cells (henceforth CD8.4 F5) from unimmunized animals. The F5 TCR is specific for influenza NP68 and has been shown to have a very low level of self-reactivity (Ge *et al*, 2004; Hogan *et al*, 2013). We observed that CD8.4 F5 T cells had lower levels of CD8 and TCR and elevated levels of CD5 and IL-7R in comparison with CD8WT F5 T cells (Fig EV1A). Because the downregulation of CD8 and expression of CD5 and IL-7R correlate with the intensity of homeostatic TCR signaling (Park *et al*, 2007), we could conclude that CD8.4 indeed enhances homeostatic TCR signaling. However, we did not detect upregulation of memory markers, CD44, CD122, and LFA-1 on CD8.4 F5 T cells (Figs 1A and EV1A). CD8.4 F5 T cells showed slightly stronger antigenic responses, measured as CD25 and CD69 upregulation, than CD8WT F5 T cells *in vitro* upon activation with the cognate antigen, NP68, or a lower affinity antigen, NP372E (Shotton & Attaran, 1998; Fig 1B). Accordingly, CD8.4 F5 T cells expanded more than CD8WT F5 T cells after the immunization with NP68 peptide (Fig 1C). Infection with transgenic *Listeria monocytogenes* expressing NP68 (Lm-NP68) induced stronger expansion and formation of larger KLRG1$^+$IL-7R$^-$ short-lived effectors and KLRG1$^-$IL-7R$^+$ memory-precursor subsets in CD8.4 F5 than in CD8WT F5 T cells (Figs 1D and EV1B). Collectively, these data showed that CD8-Lck coupling frequency sets the sensitivity of peripheral T cells to self-antigens during homeostasis and to foreign cognate antigens during infection. However, supraphysiological CD8-Lck coupling in CD8.4 F5 T cells does not induce differentiation into memory-phenotype T cells in unimmunized mice.

Whereas the level of self-reactivity of F5 T cells is very low, transgenic OT-I T cells, specific for chicken ovalbumin (OVA), exhibit a relatively high level of self-reactivity (Ge *et al*, 2004; Hogan *et al*, 2013). We tested whether a combination of a relatively highly self-reactive OT-I TCR and supraphysiological CD8-Lck coupling is sufficient to induce VM T cells. For this reason, we compared monoclonal OT-I Rag$^{-/-}$ T cells (henceforth CD8WT OT-I) and CD8.4 knock-in OT-I Rag$^{-/-}$ T cells (henceforth CD8.4 OT-I) from unimmunized animals. As expected, CD8.4 OT-I T cells exhibited signs of stronger homeostatic TCR signaling than CD8WT OT-I

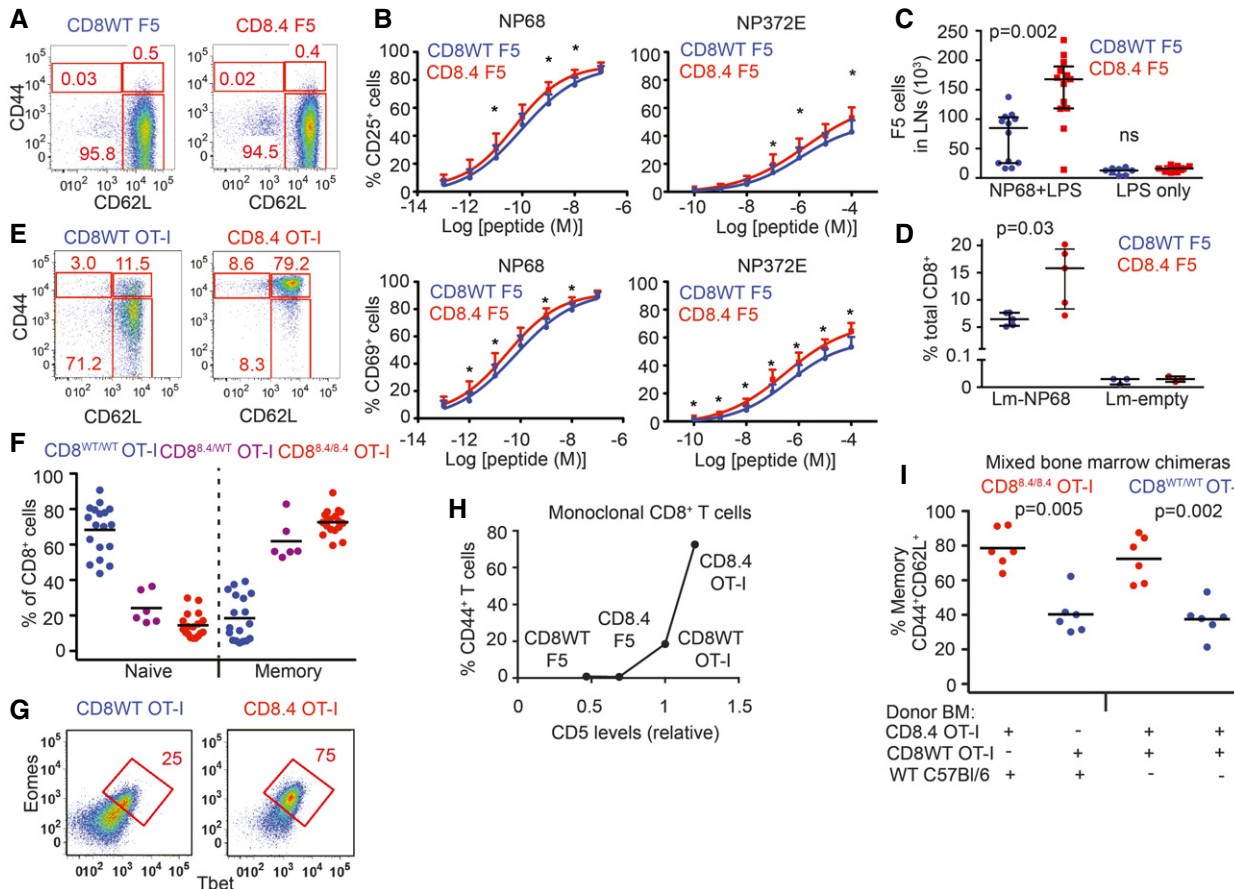

**Figure 1.  Supraphysiological CD8-Lck coupling induces differentiation into VM T cells in a clone-specific manner.**

A     LN cells isolated from CD8WT F5 and CD8.4 F5 mice were analyzed by flow cytometry (gated as viable CD8$^+$CD4$^-$). A representative experiment out of 4 in total.

B     LN cells isolated from CD8WT F5 and CD8.4 F5 mice were stimulated with antigen-loaded DCs overnight and CD69 or CD25 expression was analyzed by flow cytometry. Mean + SEM. n = 7 independent experiments. *P < 0.05.

C     2 × 10$^6$ CD8WT F5 or CD8.4 F5 LN T cells were adoptively transferred into Ly5.1 WT host 1 day prior to immunization with NP68 peptide and LPS or LPS only. Three days after the immunization, donor Ly5.2$^+$ Ly5.1$^-$ CD8$^+$ T cells from LN of the host mice were analyzed by flow cytometry and counted. Median ± interquartile range. n = 8–13 mice from seven independent experiments.

D     1 × 10$^5$ CD8WT F5 or CD8.4 F5 LN T cells were adoptively transferred into Ly5.1 WT hosts and immunized with WT Lm (empty) or Lm-NP68. Six days after the immunization, the percentage of donor Ly5.2$^+$ Ly5.1$^-$ CD8$^+$ T cells among all CD8$^+$ T cells was determined. Median ± interquartile range. n = 3–5 mice from three independent experiments.

E, F   LN cells isolated from CD8WT OT-I, CD8.4 OT-I, and heterozygous CD8$^{WT/8.4}$ mice were analyzed by flow cytometry (gated as viable CD8$^+$ CD4$^-$). Percentages of naïve (CD44$^-$ CD62L$^+$) and memory (CD44$^-$ CD62L$^+$) CD8$^+$ T cells were determined. n = 6–18 mice per group from at least five independent experiments.

G     Expression of Eomes and Tbet in CD8$^+$ LN T cells isolated from CD8WT OT-I and CD8.4 OT-I mice was determined by flow cytometry. A representative experiment out of 3 in total.

H     Relationship between relative CD5 levels (CD5 on CD8WT OT-I was arbitrarily set as 1) percentage of VM T cells (CD44$^+$ CD62L$^+$) using the data from indicated TCR transgenic strains. Mean value of n = 5–8 mice per group from at least five independent experiments.

I     Irradiated Rag2$^{-/-}$ host mice were transplanted with B-cell and T-cell depleted bone marrow from Ly5.1 C57Bl/6 together with CD8.4 OT-I or CD8WT OT-I bone marrow in 1:1 ratio (first two datasets) or, with bone marrow from CD8.4 OT-I and CD8WT OT-I mice in 1:1 ratio (last two datasets). Eight weeks later, LN cells were isolated and analyzed by flow cytometry. n = 6 mice per group from three independent experiments.

Data information: Statistical significance was determined using two-tailed Wilcoxon signed-rank test (B) and Mann–Whitney test (C, D, I).
Source data are available online for this figure.

T cells, including downregulation of TCR, CD8, and increased levels of CD5 and CD127 (Fig EV1C and D). Interestingly, the majority of the CD8.4 OT-I T cells exhibited CM phenotype including CD44$^+$ CD62L$^+$ double positivity, increased levels of CD122 and LFA-1, increased forward-scatter signal, and expression of transcription factors Tbet and Eomes (Figs 1E–G and EV1C), which was in a striking contrast with analogical experiments with CD8WT/CD8.4 F5 T

cells (Figs 1A and EV1A). The CD8$^{WT/CD8.4}$ heterozygous OT-I T cells showed an intermediate frequency of memory T cells (Fig 1F). Because CD8.4 OT-I T cells exhibited features of memory T cells without encountering their foreign cognate antigen, we concluded that CD8.4 induced a differentiation of OT-I T cells into VM cells. When we compared monoclonal T cells from all four transgenic mouse strains tested, we identified a relationship between surface

levels of CD5, a commonly used marker of self-reactivity, and the frequency of VM T-cell formation. However, the dependency was not linear, but showed a threshold behavior, indicating that only the most self-reactive T cells have the potential to develop into VM T cells (Figs 1H and EV1E).

To address whether CD8.4 induces VM T-cell formation in OT-I T cells in a T-cell intrinsic manner, we generated mixed bone marrow chimeric animals where both CD8.4 and CD8WT populations were present. We transplanted bone marrow cells from Ly5.1 WT mouse together with bone marrow cells from either Ly5.2 CD8WT OT-I or Ly5.2 CD8.4 OT-I mice into an irradiated Rag2$^{-/-}$ recipient. CD8.4 OT-I T cells generated substantially more VM T cells than CD8WT OT-I T cells (Fig 1I). In an alternative setup, we cotransferred mixed bone marrow cells from Ly5.1 CD8WT OT-I and Ly5.2 CD8.4 OT-I animals into an irradiated Rag2$^{-/-}$ recipient to compare these two subsets in a single mouse. Again, CD8.4 OT-I T cells generated significantly more VM T cells than CD8WT OT-I T cells (Fig 1I). These experiments showed that CD8.4 OT-I T cells intrinsically trigger the memory differentiation program.

## CD8-Lck coupling frequency regulates the size of virtual memory compartment in polyclonal repertoire

In a next step, we addressed how the CD8-Lck stoichiometry regulates the size of virtual memory compartment in a polyclonal T-cell pool. Interestingly, CD8.4 polyclonal mice showed significantly higher frequency of VM CD8$^+$ T cells than CD8WT control animals, although most CD8.4 T cells still showed a naïve phenotype (Fig 2A and B). Thus, CD8.4 induced the VM T-cell formation only in a subset of polyclonal CD8$^+$ T cells, implying that enhanced CD8-Lck coupling has clone-specific effects. The VM T cells from both CD8WT and CD8.4 mice rapidly produced IFNγ after stimulation with PMA/ionomycin, showing that CD8WT and CD8.4 VM T cells are indistinguishable in this functional trait, typical for memory T cells (Fig EV2A). Importantly, the analysis of mice in germ-free condition showed elevated frequency of CD44$^+$ and Tbet/Eomes double-positive T cells in CD8.4 mice when compared to CD8WT (Fig 2C and D), demonstrating that supraphysiological CD8-Lck coupling indeed promotes formation of VM T cells independently of the exposure to foreign antigens.

Although the VM and true CM T cells are very similar in many aspects, VM T cells were previously reported to express lower levels of CD49d and slightly higher levels of CD122 than true CM T cells (Haluszczak et al, 2009; Chiu et al, 2013; Sosinowski et al, 2013; White et al, 2016). However, CD49d as a marker discriminating VM and true CM T cells has not been validated using T cells from germ-free animals. For the first time, we could show that CD49d$^-$ and CD49d$^+$ T cells within the CD44$^+$ compartment of SPF and germ-free mice occur at comparable frequencies (Fig 2E). Only the CD49d$^-$ CD44$^+$, but not the CD49d$^+$ CD44$^+$, subset was expanded in the CD8.4 mouse, indicating that these subsets are not related (Fig 2E). Accordingly, CD122$^{HI}$ CD49d$^-$ CD44$^+$ T cells, but not CD122$^{LOW}$ CD49d$^+$ CD44$^+$ T cells, were more abundant in germ-free CD8.4 mouse than in germ-free CD8WT mouse (Figs 2F and G, and EV2B). Collectively, these data implied that CD122$^{HI}$ CD49d$^-$ memory T cells represent the CD8$^+$ VM T-cell population, which originates from naïve T cells with a relatively strong level of self-reactivity independently of foreign antigens.

## Virtual memory T cells use distinct TCR repertoire than naïve T cells

Based on the previous data, generation of VM T cells can be understood as a fate decision checkpoint of individual naïve T cells (staying naïve vs. becoming VM), where the decision is based on the level of self-reactivity of the T cell's TCR. This hypothesis predicts that naïve and VM T-cell compartments should contain different T-cell clones with distinct TCR repertoires. We analyzed the TCR repertoires by using a Vβ5 transgenic mouse with fixed TCRβ from the OT-I TCR (Fink et al, 1992). The advantage of this mouse is that it generates a polyclonal population of T cells, but the variability between the clones is limited to TCRα chains. Moreover, pairing of TCRα and TCRβ upon repertoire analyses is not an issue in this model. Last, but not least, this mouse has a relatively high frequency of oligoclonal T cells that recognize the ovalbumin antigen (Zehn & Bevan, 2006).

Unimmunized Vβ5 mice have a frequency of memory CD8$^+$ T cells around 10–15% which is comparable to wild-type mice (Fig 3A). When we analyzed the frequency of VM vs. naïve T cells among particular T-cell subsets defined by the expression of particular TCRVα segments, we observed that TCRVα3.2$^+$ T cells are comparable to the overall population, TCRVα2$^+$ T cells are slightly enriched for the VM T cells, and TCRVα8.3$^+$ T cells have lower frequency of VM T cells than the overall population (Fig 3A). These data suggested that naïve and VM T cells might have distinct TCR repertoires. However, the differences between the subsets were only minor, most likely because particular TCRVα subsets had still significant intrinsic diversity. To further reduce clonality in our groups, we gated on K$^b$-OVA-specific TCRVα3.2$^+$, TCRVα2$^+$, and TCRVα8.3$^+$ T cells (Fig 3B). Interestingly, around 50% of OVA-specific TCRVα2$^+$ clones exhibited VM phenotype, whereas vast majority of OVA-specific TCRVα8.3$^+$ T cells were naïve and OVA-specific TCRVα3.2$^+$ T cells had intermediate frequency of VM T cells in peripheral LN, mesenteric LN as well as in the spleen (Fig 3B and C). Accordingly, the frequency of TCRVα2$^+$ T cells is almost 10-fold higher in VM than in naïve OVA-specific T-cell population, whereas the frequency of TCRVα8.3$^+$ T cells is slightly lower in VM than in naïve OVA-specific T cells (Fig EV3A). Similar differences between total and OVA-specific TCRVα3.2$^+$, TCRVα2$^+$, and TCRVα8.3$^+$ CD8$^+$ T cell subsets were observed in germ-free Vβ5 mice (Fig 3D and E), confirming the VM identity of memory-phenotype T cells in the Vβ5 mice. In addition, OVA-specific TCRVα2$^+$ had higher levels of CD5 than TCRVα8.3$^+$ (Fig EV3B and C), suggesting that OVA-specific TCRVα2$^+$ clones are on average more self-reactive than TCRVα8.3$^+$ clones. This explains why more OVA-specific TCRVα2$^+$ T cells than TCRVα8.3$^+$ T cells acquire the VM phenotype in Vβ5 mice.

Based on the analysis of particular TCRVα subsets, we hypothesized that naïve and VM T cells would contain different TCR clonotypes. We cloned and sequenced genes encoding for TCRα chains from OVA-reactive CD8$^+$ VM and naïve T-cell subsets from germ-free Vβ5 mice using primers specific for TRAV14 (TCRVα2) and TRAV12 (TCRVα8) TCR genes (Table EV1). The distribution of the clonotypes as well as TRAJ usage was significantly different between VM and naïve subsets (Figs 3F and EV3D). We observed essentially two types of clonotypes that were captured in more than 1 experiment (Fig 3F). One type of clonotypes, called "VM clones",

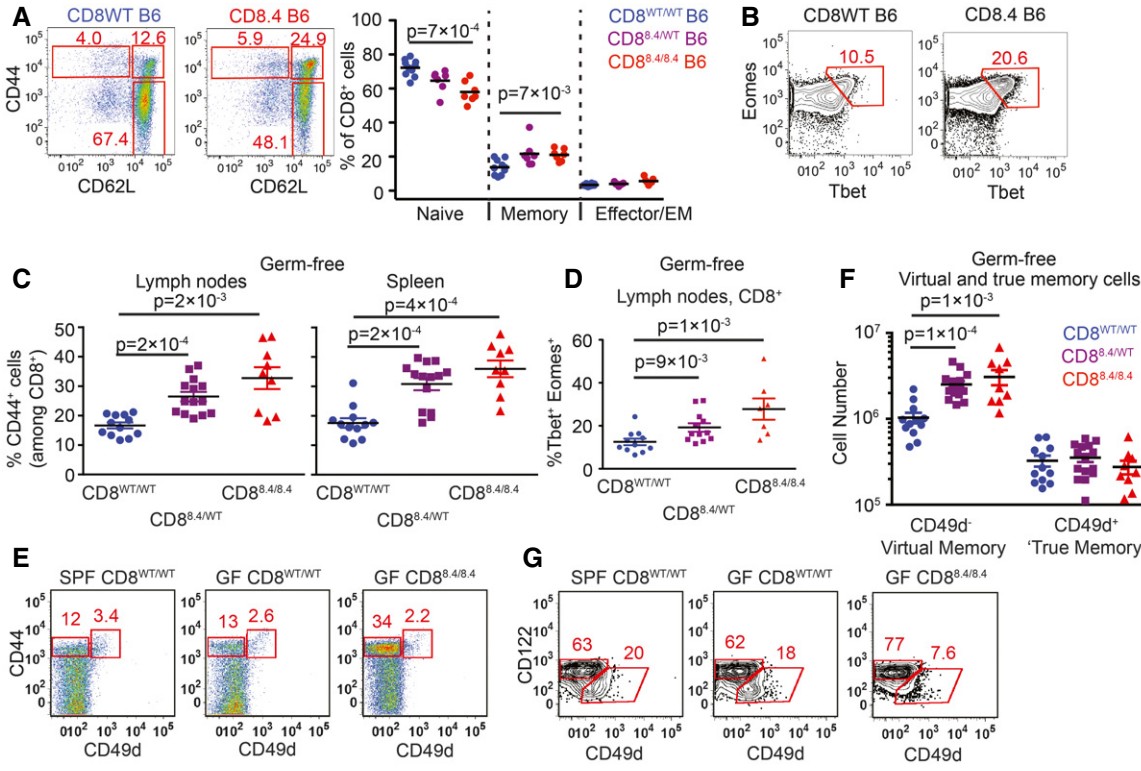

**Figure 2. CD8-Lck coupling is a limiting factor for the size of the virtual memory T-cell compartment.**

A, B    Percentages of naïve (CD44⁻ CD62L⁺), central memory (CD44⁺ CD62L⁺), and effector/effector memory (CD44⁺ CD62L⁻) (A) and Eomes⁺ Tbet⁺ (B) CD8⁺ LN T cells isolated from CD8WT and CD8.4 mice were determined by flow cytometry. Representative experiments out of seven (A) or five (B) in total.

C–F    LN cells and splenocytes were isolated from germ-free polyclonal CD8WT, CD8.4, and heterozygous CD8$^{WT/8.4}$ mice and CD8⁺ T cells were analyzed by flow cytometry. (C) Percentage of CD44⁺ T cells among CD8⁺ LN cells and splenocytes. Mean + SEM. *n* = 9–14 mice per group from four independent experiments. (D) Percentage of Tbet⁺ Eomes⁺ double-positive T cells. Mean ± SEM. *n* = 7–12 mice per group from three independent experiments. (E) Percentage of CD44⁺ CD49d⁻ VM and CD44⁺ CD49d⁺ true memory T cells in the spleen. A representative experiment out of four in total. (F) Absolute numbers of CD8⁺ CD44⁺ CD49d⁻ VM and CD8⁺ CD44⁺ CD49d⁺ true memory T cells in LN and the spleen were quantified. Mean + SEM. *n* = 9–14 mice from four independent experiments.

G    Percentage of CD122$^{HI}$ CD49d⁻ VM and CD122$^{LOW}$ CD49⁺ true CM cells among CD8⁺ CD44⁺ CD62L⁺ CM T cells isolated from LN. A representative experiment out of three in total.

Data information: Statistical significance was determined using two-tailed Mann–Whitney test.
Source data are available online for this figure.

was enriched among VM T cells and was also present in naïve T cells. The other type, "naïve clones", was almost exclusively detected in naïve T cells. These data demonstrate that naïve and VM T-cell population contain different T-cell clones.

To directly investigate whether TCR is the main factor that determines whether a particular T cell has the potential to differentiate into VM T cells, we established a protocol to generate monoclonal T-cell populations using transduction of a particular TCRα-encoding gene in a retrogenic vector into immortalized hematopoietic stem cells (Ruedl *et al*, 2008). We transduced immortalized Vβ5 Rag2$^{-/-}$ hematopoietic stem cells with expression vectors encoding for two VM TCRα clones (V14-C1 and V14-C2), three naïve TCRα clones (V14-C6, V14-C7, and V14-C17), or OT-I TCRα as a control (Fig EV3E). At least 8 weeks after the transplantation of the progenitors into a Ly5.1 recipient, we analyzed the cell fate of the donor monoclonal T-cell populations. T cells expressing VM TCR clones formed a significant VM T-cell population, whereas T cells expressing naïve TCR clones formed a homogenous naïve population (Figs 3G and EV3F). These data demonstrate that the virtual

memory T cells are formed only from certain T-cell clones and that the TCR sequence determines whether a T cell differentiates into a VM T cell or stays naïve.

In a next step, we addressed whether "VM clones" are more self-reactive than "naïve clones". We compared levels of CD5, a commonly used reporter for self-reactivity, on naive (CD44⁻) populations of retrogenic monoclonal T cells. "VM clones" expressed significantly higher CD5 levels than "naïve clones", indicating that "VM clones" are indeed T cells with a relatively high level of self-reactivity (Figs 3H and EV3G). Interestingly, retrogenic OT-I T cells represented an intermediate VM population, which corresponds to their level of self-reactivity (Figs 1H, 3G, and EV3F and G). Moreover, the relative size of retrogenic OT-I VM population well corresponded to the frequency of VM T cells in conventional transgenic OT-I TCR mice (Fig 1E and H), indicating that the protocol for generation of retrogenic monoclonal T cells itself does not have a strong effect on VM formation. Overall, these results document that only relatively highly self-reactive clones form VM T cells.

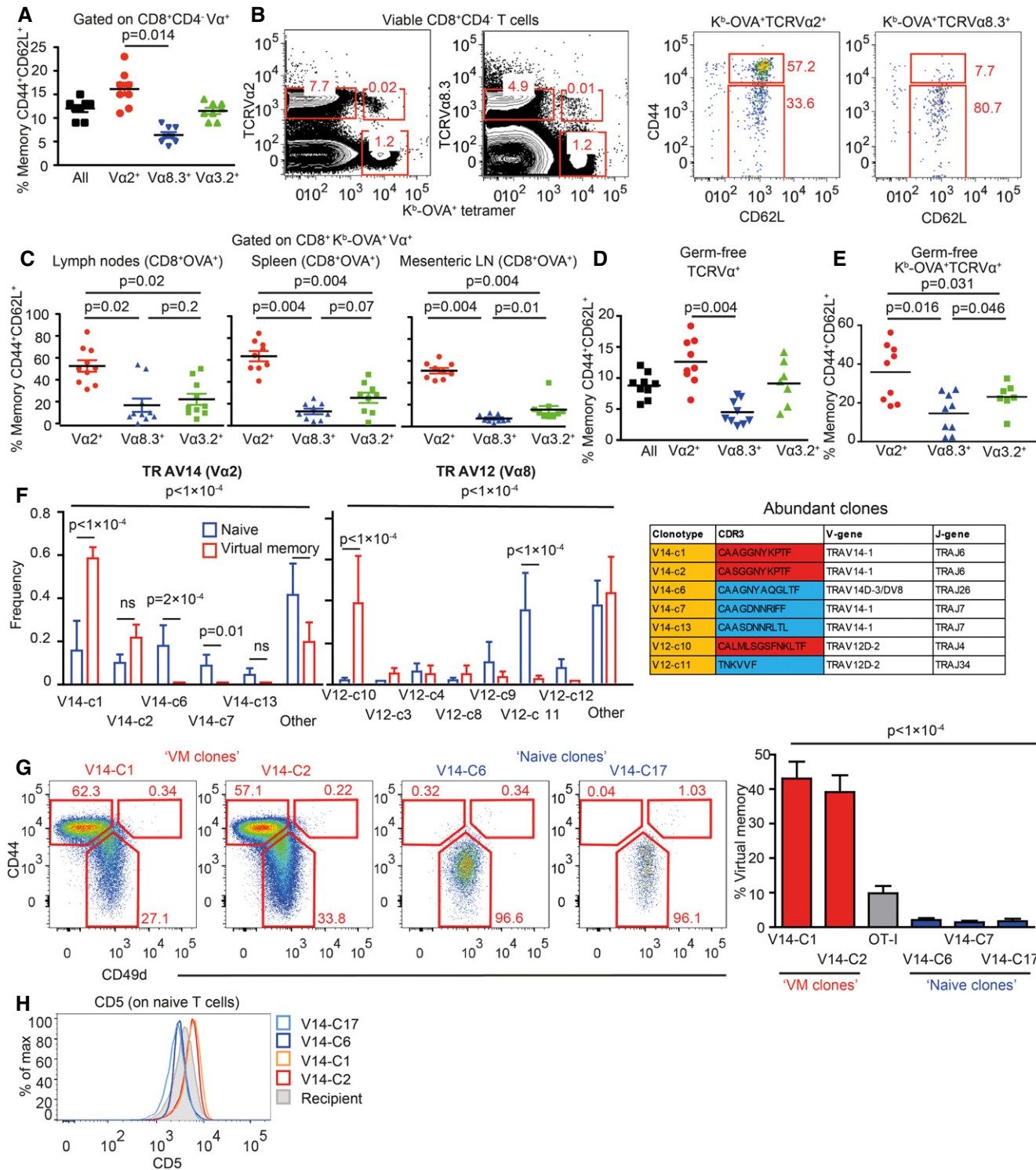

Figure 3.

## Virtual memory T cells represent an intermediate stage between naïve and memory T cells

The relationship of the differentiation programs of naïve, VM, and true CM T cells is unclear. So far, CD49d and, to a lesser extent, CD122 were the only known markers discriminating between true

CM memory and VM T cells (Haluszczak *et al*, 2009; Chiu *et al*, 2013; Sosinowski *et al*, 2013; White *et al*, 2016). To compare their gene expression profiles, we performed deep RNA sequencing of the transcripts from sorted CD8$^+$ naïve (CD44$^-$ CD62L$^+$) and VM (CD44$^+$ CD62L$^+$ CD49d$^-$) T cells isolated from germ-free animals and from true antigen-specific CM T cells (TM) (K$^b$-OVA$^+$ CD44$^+$

◄

**Figure 3.  Virtual memory and naïve T cells use different TCR repertoires.**

A    LN cells isolated from Vβ5 mice were stained for CD8, CD4, CD44, CD62L, and Vα2 or Vα8.3 or Vα3.2. CD8$^+$ T cells were gated as CD8$^+$ CD4$^-$ and then the percentage of CD44$^+$ CD62L$^+$ memory T cells among CD8$^+$ Vα2$^+$ or CD8$^+$ Vα8.3$^+$ or CD8$^+$ Vα3.2$^+$ T cells was determined by flow cytometry. Mean, $n$ = 8 mice from eight independent experiments.

B, C    Cells isolated from peripheral LN (B, C), mesenteric LN (C), and the spleen (C) were stained as in (A) with the addition of OVA tetramer. The OVA-reactive Vα-specific CD8$^+$ T cells were gated and the percentage of CD44$^+$ CD62L$^+$ memory T cells was determined by flow cytometry. $n$ = 9–10 mice from five independent experiments.

D    The same experiment as in (A) was performed using germ-free Vβ5 mice. Mean, $n$ = 7–9 mice from four independent experiments.

E    The same experiment as in (B, C) was performed using a mixture of T cells isolated from LNs and the spleen from germ-free Vβ5 mice. Mean, $n$ = 7–9 from 2 to 3 independent experiments.

F    RNA was isolated from memory (CD44$^+$CD62L$^+$) and (CD44$^+$CD62L$^+$) K$^b$-OVA$^+$ 4mer$^+$ T cells sorted from LNs and the spleen of germ-free Vβ5 mice. TCRα encoding genes using either TRAV12 (corresponding to Vα8) or TRAV14 (corresponding to Vα2) were cloned and sequenced. 12–20 clones were sequenced in each group/experiment. Clonotypes identified in at least two experiments are shown. Mean frequency + SEM, $n$ = 4 independent experiments. Statistical significance was determined by chi-square test (global test) and paired two-tailed $t$-tests as a post-test (individual clones). CDR3 sequences of clonotypes enriched in naïve or VM compartments are shown in the table.

G, H    Retroviral vectors encoding selected TCRα clones were transduced into immortalized Rag2$^{-/-}$ Vβ5 bone marrow stem cells. These cells were transplanted into an irradiated Ly5.1 recipient. (G) At least 8 weeks after the transplantation, frequency of virtual memory T cells among LN donor T cells (CD45.2$^+$ CD45.1$^-$ GFP$^+$) was analyzed. Mean + SEM; $n$ = 10–21 mice from 2 to 7 independent experiments. Statistical significance was tested using Kruskal–Wallis test. (H) CD5 levels on naïve monoclonal T cells were detected by flow cytometry. Representative mice out of 9–14 in total from two to four independent experiments.

Data information: (A, C–E) Statistical significance was determined by two-tailed Wilcoxon signed-rank test.
Source data are available online for this figure.

CD62L$^+$), generated by infecting Vβ5 mice with Lm expressing OVA (Lm-OVA). The data showed that naïve, VM, and TM T cells represent three distinct T-cell populations (Fig EV4A and B). We identified genes differently expressed in VM T cells in comparison with naïve or TM T cells (Tables EV2–EV5). Based on previously published data (Kaech *et al*, 2002; Luckey *et al*, 2006; Wherry *et al*, 2007), we established lists of memory signature and naïve signature genes. As expected, memory signature genes were enriched in TM T cells and naïve signature genes were enriched in naïve T cells (Figs 4A and EV4C). Interestingly, VM T cells exhibited an intermediate gene expression profile (Figs 4A and EV4C). Pairwise rotation gene set tests revealed the hierarchy in the enrichment for memory signature genes and for naïve signature genes as TM > VM > naïve, and naïve > VM > TM, respectively (Fig 4B). VM T cells also showed an intermediate expression of cytokine and chemokine encoding genes (Figs 4C and EV4D). The transcription of genes encoding for cytokine and chemokine receptors in VM T cells seemed to be also somewhere half-way between naïve and true CM T cells (Fig EV4E and F).

We further investigated selected differentially expressed genes between VM and true CM cells on a protein level. RNA encoding for CX3CR1 and NRP1 showed enrichment in TM vs. VM T cells. For this reason, we compared surface levels of CX3CR1 and NRP1 on naïve and VM T cells from unprimed mice and on TM and effector/effector memory T cells from LM-OVA infected mouse during the memory phase (Fig 4D and E). In contrast to VM T cells, a significant percentage of effector and TM T cells expressed CX3CR1 and NRP1, confirming the transcriptomic data and suggesting that VM T cells can be characterized as CX3CR1 and NRP1 negative.

## Differentiation into virtual memory T cells does not break self-tolerance

VM T cells provide better protection against Lm than naïve T cells and respond to proinflammatory cytokines IL-12 and IL-18 by producing IFNγ (Haluszczak *et al*, 2009; Lee *et al*, 2013; White *et al*, 2016). Moreover, VM T cells express higher levels of several killer lectin-like receptors than naïve T cells (Table EV4 and White *et al*,

2016). The combination of a hyperresponsive differentiation state with the expression of highly self-reactive TCRs suggests that VM CD8$^+$ T cells could be less self-tolerant than naïve T cells and might represent a risk for inducing autoimmunity. We used CD8WT OT-I T cells (mostly naïve) and CD8.4 OT-I T cells (mostly VM) for a functional comparison of naïve and VM T cells with the same TCR specificity. VM CD8.4 OT-I T cells, but not naïve OT-I T cells, rapidly produced IFNγ after the stimulation by PMA/ionomycin or cognate antigen (Fig 5A), supporting the idea of an autoimmune potential of VM T cells. We directly tested this hypothesis using an experimental model of autoimmune diabetes (King *et al*, 2012). We transferred CD8WT OT-I or CD8.4 OT-I T cells into RIP.OVA mice expressing OVA under the control of rat insulin promoter (Kurts *et al*, 1998) and primed them with Lm-OVA or Lm-Q4H7 (King *et al*, 2012). Q4H7 is an antigen that binds to the OT-I TCR with a low affinity and does not negatively select OT-I T cells in the thymus (Daniels *et al*, 2006; Stepanek *et al*, 2014). Thus, Q4H7 resembles a self-antigen that positively selected peripheral T cells might encounter at the periphery. Surprisingly, CD8.4 OT-I T cells were not more efficient in inducing the autoimmune diabetes than CD8WT OT-I T cells in any tested condition (Figs 5B and EV5A). Adoptively transferred naïve OT-I, CD8.4 OT-I, and even TM OT-I T cells did not promote clearance of Lm-Q4H7 in this experimental setup (relatively low number of injected CFUs, low antigen affinity; Fig EV5B), excluding the possibility that the bacterial burden differs between experimental groups.

To further analyze the functional responses of VM T cells, we stimulated CD8.4 OT-I and CD8WT OT-I T cells with dendritic cells loaded with OVA or suboptimal cognate antigens T4 or Q4H7 *ex vivo*. No significant difference in the upregulation of CD69 or CD25 between naïve and VM cells was observed in the case of high-affinity OVA stimulation. However, the responses to antigens with suboptimal affinity to the TCR differed between these two cell types. Interestingly, although CD8.4 OT-I T cells showed stronger CD69 upregulation than naïve T cells when stimulated with low antigen dose, when the antigen dose was high, the response of VM T cells was lower than that of naïve T cells (Fig 5C). Upregulation of CD25 was lower in CD8.4 OT-I T cells than in naïve T cells, when activated with the low-affinity antigens (Fig 5D).

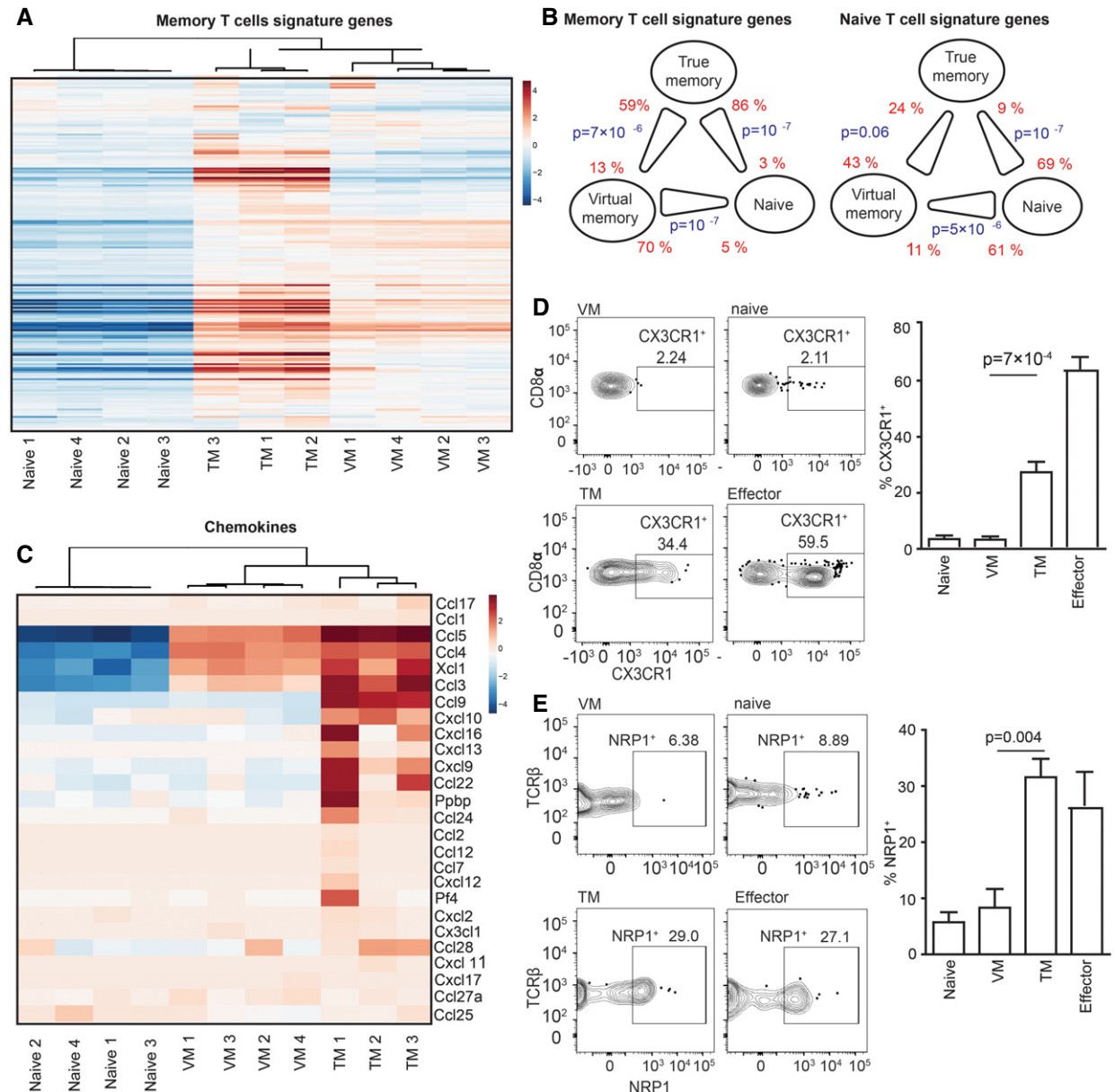

**Figure 4. Virtual memory T cells represent an intermediate stage between naïve and true memory T cells.**

A–C  Transcriptomes of naïve (*n* = 4), VM (*n* = 4), and TM (*n* = 3) CD8$^+$ T cells were analyzed by deep RNA sequencing. (A) Enrichment of CD8$^+$ memory signature genes (as revealed by previous studies) in naïve, virtual memory, and true memory T cells. (B) Pairwise comparisons between naïve, VM, and TM CD8$^+$ cells for the overall enrichment of the memory signature and naïve signature gene sets by a method ROAST. The thick end of the connecting line between the populations indicates the population with the overall relative enrichment of the gene set, the percentage of the genes from the gene set that are more expressed in the indicated population (z-score > sqrt(2)) over the opposite population is indicated. (C) The relative enrichment of chemokine encoding transcripts in the samples is shown.

D, E  Surface staining for CX3CR1 (D) and NRP1 (E) was performed on naïve (gated as CD62L$^+$CD44$^-$CD49d$^{low}$) and VM (CD62L$^+$CD44$^+$CD49d$^{low}$) K$^b$-OVA-4mer$^+$ CD8$^+$ T cells isolated from unprimed Vβ5 mouse and on true CM memory (gated as CD62L$^+$CD44$^+$CD49d$^{high}$) and effector/effector memory (CD62L$^-$CD44$^+$CD49d$^{high}$) K$^b$-OVA-4mer$^+$ CD8$^+$ T cells isolated from Vβ5 mouse 30–45 days after Lm-OVA infection. A representative experiment out of five (D) or four (E) in total is shown. Mean percentage + SEM of CX3CR1$^+$ and NRP1$^+$ cells within the particular population is shown. (D) *n* = 10 immunized mice and five unprimed mice from five independent experiments. (E) *n* = 8 immunized mice and four unprimed mice from four independent experiments. Statistical analysis was performed by two-tailed Mann–Whitney test.

Source data are available online for this figure.

Next, we examined antigenic responses of naïve and VM T cells *in vivo*. CD8WT and CD8.4 OT-I T cells show comparable expansion when primed by Lm-OVA or Lm-Q4H7 (Fig EV5C). However, CD8.4 OT-I formed significantly less memory precursors (IL-7R$^+$KLRG1$^-$), more IL-7R$^+$KLRG1$^+$ double-positive cells, and slightly more short-lived effector cells (IL-7R$^-$KLRG1$^+$) than CD8WT OT-I T cells upon

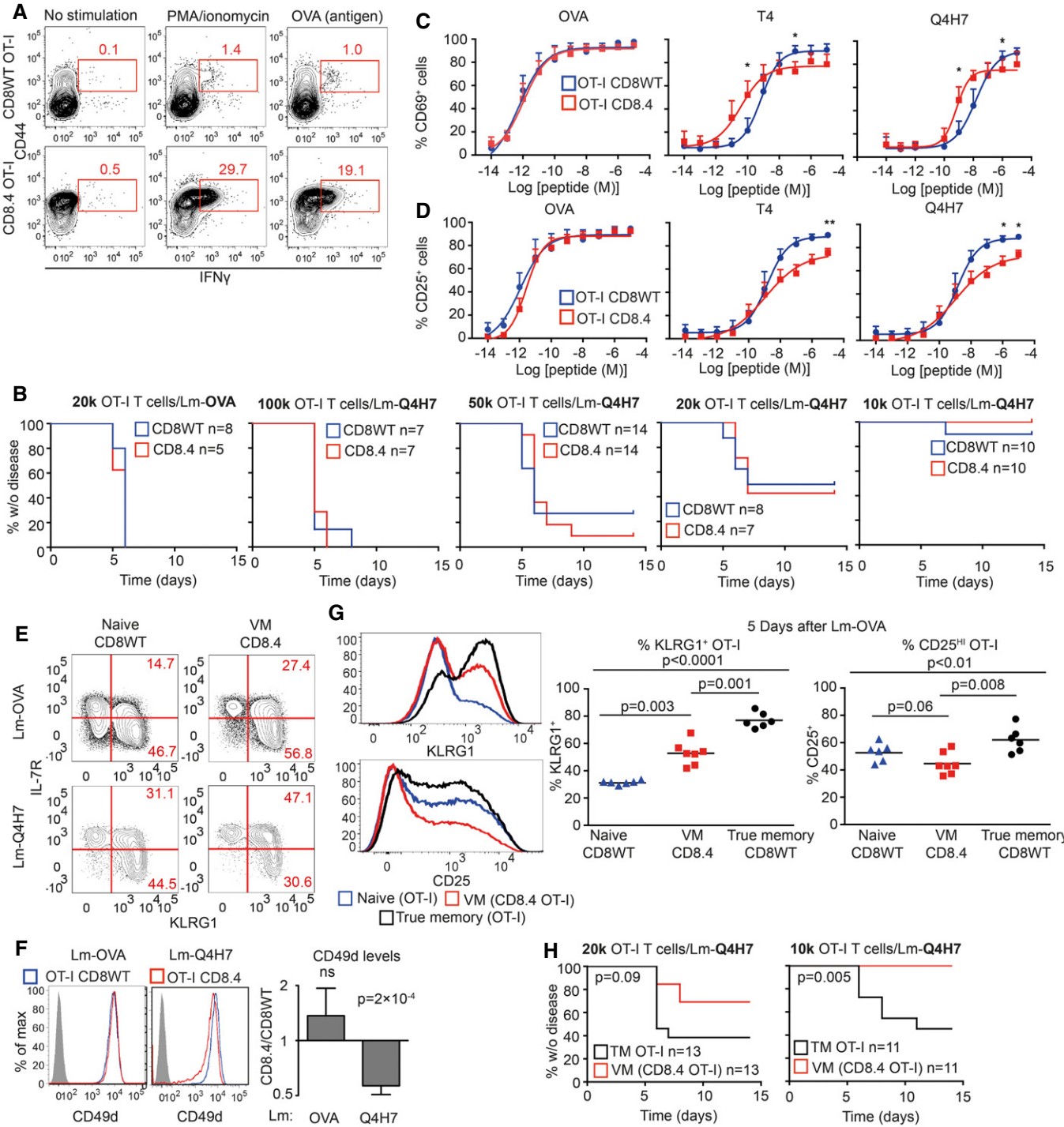

**Figure 5.**

priming with Lm-OVA (Figs 5E and EV5C). Interestingly, upon priming with low-affinity Lm-Q4H7, VM T cells formed less short-lived effector cells and more IL-7R$^+$KLRG1$^+$ double-positive cells (Figs 5E and EV5C). Moreover, CD8.4 OT-I T cells did not upregulate CD49d (a subunit of VLA-4) to the same extent as CD8WT OT-I upon immunization with Lm-Q4H7 (Fig 5F). Collectively, VM T cells exhibited several signs of hyporesponsiveness in comparison with naïve T cells upon low-affinity antigenic stimulation: lower

upregulation of CD25 *in vitro*, lower frequency of short-lived effector cells, and lower CD49d upregulation.

Subsequently, we compared responses of VM CD8.4 T cells to naïve and true memory OT-I T cells *in vivo*. True memory showed stronger upregulation of KLRG1 and CD25 than VM T cells upon Lm-OVA challenge (Fig 5G). Importantly, true memory OT-I T cells were more potent in inducing the autoimmune diabetes in our RIP.OVA model (Figs 5H and EV5D). Collectively, these data

◀

**Figure 5.  Virtual memory T cells are as self-tolerant as naïve T cells.**

A    LN cells isolated from CD8WT OT-I and CD8.4 OT-I mice were stimulated with PMA and ionomycin or OVA peptide in the presence of BD GolgiStop for 5 h and the production of IFNγ was analyzed by flow cytometry (gated as CD8+). A representative experiment of four in total.

B    Indicated number of CD8WT OT-I or CD8.4 OT-I T cells were adoptively transferred into RIP.OVA hosts, which were infected with Lm-OVA or -Q4H7 1 day later. The glucose in the urine was monitored for 14 days. The percentage of non-diabetic mice in time is shown. Differences between OTI-I and CD8.4 were not significant by log-rank test. *n* = 5–11 (indicated) mice per group in 3–6 independent experiments.

C, D    CD8WT OT-I or CD8.4 OT-I LN T cells were stimulated *ex vivo* with dendritic cells loaded with varying concentrations of OVA, Q4R7, Q4H7 peptides overnight and the expression of CD69 (C) and CD25 (D) on CD8+ T cells was analyzed. Mean + SEM. *n* = 3–5 independent experiments. Statistical significance was determined paired two-tailed Student's *t*-test (C, D). *$P < 0.05$, **$P < 0.01$.

E, F    CD8WT OT-I or CD8.4 OT-I LN T cells were adoptively transferred to polyclonal Ly5.1 host mice, which were infected 1 day later with transgenic Lm-OVA or -Q4H7. Six days after the infection, splenocytes from the hosts were isolated and analyzed for the expression of IL-7R, KLRG1 (E) or CD49d (F). (F) A representative experiment out of four (7–9 mice per group in total). (E) *n* = 7 (Lm-OVA) or 9 (Lm-Q4H7) from four independent experiments. Statistical significance was determined using a one-value two-tailed *t*-test (for the ratio of CD49d MFI between the subsets). Normality of the data was tested using Shapiro–Wilk normality test (passing threshold $P < 0.01$).

G    $1 \times 10^4$ naïve CD8WT OT-I, VM CD8.4 OT-I, or true memory OT-I T cells loaded with 5 μM CellTrace Violet were injected into Ly5.1 recipients followed by immunization with Lm-OVA 1 day later. Five days after the immunization, splenocytes were isolated and donor cells were examined for the expression of KLRG1 and CD25 by flow cytometry. *n* = 6 from three independent experiments. Statistical significance was performed by Kruskal–Wallis test, and selected pairs of groups were compared by Mann–Whitney test.

H    Indicated number of CD8.4 OT-I or true memory OT-I T cells were adoptively transferred into RIP.OVA hosts, which were infected with Lm-Q4H7 1 day later. The glucose in the urine was monitored for 14 days. The percentage of non-diabetic mice in time is shown. Statistical significance was calculated by log-rank test. *n* = 11 ($1 \times 10^4$ transferred cells) or 13 ($2 \times 10^4$ transferred cells) mice per group in four independent experiments.

Source data are available online for this figure.

suggest that virtual memory T cells are less efficient in their responses to the antigen *in vivo* and in inducing the autoimmune tissue pathology than true memory T cells.

We wondered whether CD8.4 OT-I T cells do respond to endogenous self-antigens Catnb and Mapk8 that were previously proposed as positive selecting antigens for OT-I T cells (Santori *et al*, 2002). In agreement with previous reports (Salmond *et al*, 2014; Oberle *et al*, 2016), we could not detect a substantial response of peripheral OT-I T cells to these antigens *in vitro* using antigen-loaded dendritic cells and *in vivo* using Lm-Catnb (Fig EV5E and F). CD8.4 OT-I T cells showed no significant response to these self-peptides as well (Fig EV5E and F). Although we could see that Lm infection induced proliferation of VM CD8.4 T cells (probably via cytokines), expression of the positive selecting self-antigen Catnb in the *Listeria* did not enhance this response at all (Fig EV5F). These experiments suggest that VM T cells are tolerant to self-antigens that have previously triggered their conversion to VM T cells.

**Retrogenic T cells as a model for functional differences between naïve and VM T cells**

To complement our data from CD8.4 OT-I VM model, we used sorted naïve and VM T cells from the OVA-specific clones V14-C1 and V14-C2 (Fig 3F–H). The advantage of this approach is that both naïve and VM express the same TCR and CD8 coreceptor and any differences between these populations can be attributed solely to their different developmental programs. We adoptively transferred these cells into RIP.OVA mice followed by infection with Lm-OVA. Naïve T cells were more efficient in inducing the autoimmune diabetes than VM T cells in case of both clones, but only the clone V14-C1 showed a statistically significant difference (Fig 6A). When we adoptively transferred naïve or VM T cells expressing V14-C1 or V14-C2 TCRs into Ly5.1 recipients followed by immunization with dendritic cells loaded with OVA or lower affinity antigen Q4R7, we observed that VM clones showed significantly lower level of upregulation of CD49d, a subunit of VLA4 important for tissue infiltration (Fig 6B). These observations were in agreement with the results

obtained with the CD8.4 OT-I model of monoclonal VM T cells, demonstrating that VM T cells are not inherently less self-tolerant than naïve T cells.

Although others and we showed that VM T cells elicit stronger responses than naïve T cells in some assays (Fig 5A and Lee *et al*, 2013), they do not show a stronger potency than naïve T cells to induce autoimmune pathology in our diabetic model. Most likely VM T cells acquire mechanisms to suppress their responses to antigens, infiltration of the tissue, and/or their effector functions. One such mechanism, contributing to the self-tolerance of VM T cells, can be lower expression of CD49d and CD25 upon activation. Collectively, these data establish that relatively strongly self-reactive T-cell clones differentiate into VM T cells and trigger a specific developmental program that enables them to efficiently response to infection, but does not increase their autoimmune potential (Fig 7).

## Discussion

We observed that CD8-Lck coupling frequency regulates intensity of TCR homeostatic signals. For the first time, we showed that the intrinsic sensitivity of the TCR signaling machinery sets the frequency of VM CD8+ T cells. We also showed that only relatively strongly self-reactive T-cell clones have the potential to form VM T cells. We identified the gene expression profile of VM T cells and showed that they represent an intermediate stage between naïve and true CM T cells. Although the combination of relatively strong self-reactivity and acquisition of the partial memory program could represent a potential risk for autoimmunity, we observed that VM T cells are not more potent than naïve T cells in a model of experimental type I diabetes.

It is well established that some memory-phenotype T cells respond to an antigen, and they have not been previously exposed to (Haluszczak *et al*, 2009; Lee *et al*, 2013; Sosinowski *et al*, 2013; Su *et al*, 2013; White *et al*, 2017). Some researchers call these cells as VM T cells and propose that they were generated in the absence of a foreign antigenic stimulation (Haluszczak *et al*, 2009; White

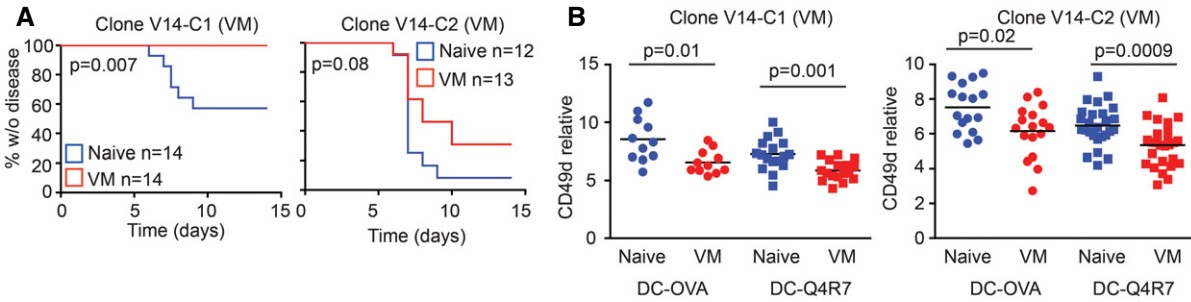

**Figure 6.  Comparison of naïve and VM subsets generated from retrogenic T-cell clones.**

A   $5 \times 10^3$ FACS-sorted naïve (CD44⁻) or VM (CD44⁺) T cells from V14-C1 or V14-C2 T cells generated via bone marrow transfer (Fig EV3E) were adoptively transferred into RIP.OVA mice followed by immunization with Lm-OVA 1 day later. Glucose concentration in the urine was monitored for 14 days. Statistical significance was tested using log-rank test. $n$ = 12–14 mice per group in 4–5 independent experiments.

B   FACS-sorted naïve (CD44⁻) or VM (CD44⁺) T cells from V14-C1 or V14-C2 T cells generated via bone marrow transfer (Fig EV3E) were adoptively transferred into RIP.OVA mice followed by immunization with OVA- or Q4R7-loaded bone marrow-derived dendritic cells. The number of adoptively transferred T cells was $1 \times 10^3$ and $3 \times 10^3$ for OVA- and Q4R7-loaded dendritic cells, respectively. $n$ = 11–27 mice per group in 5–6 independent experiments. Mean is indicated. Statistical significance was tested using Mann–Whitney test.

Source data are available online for this figure.

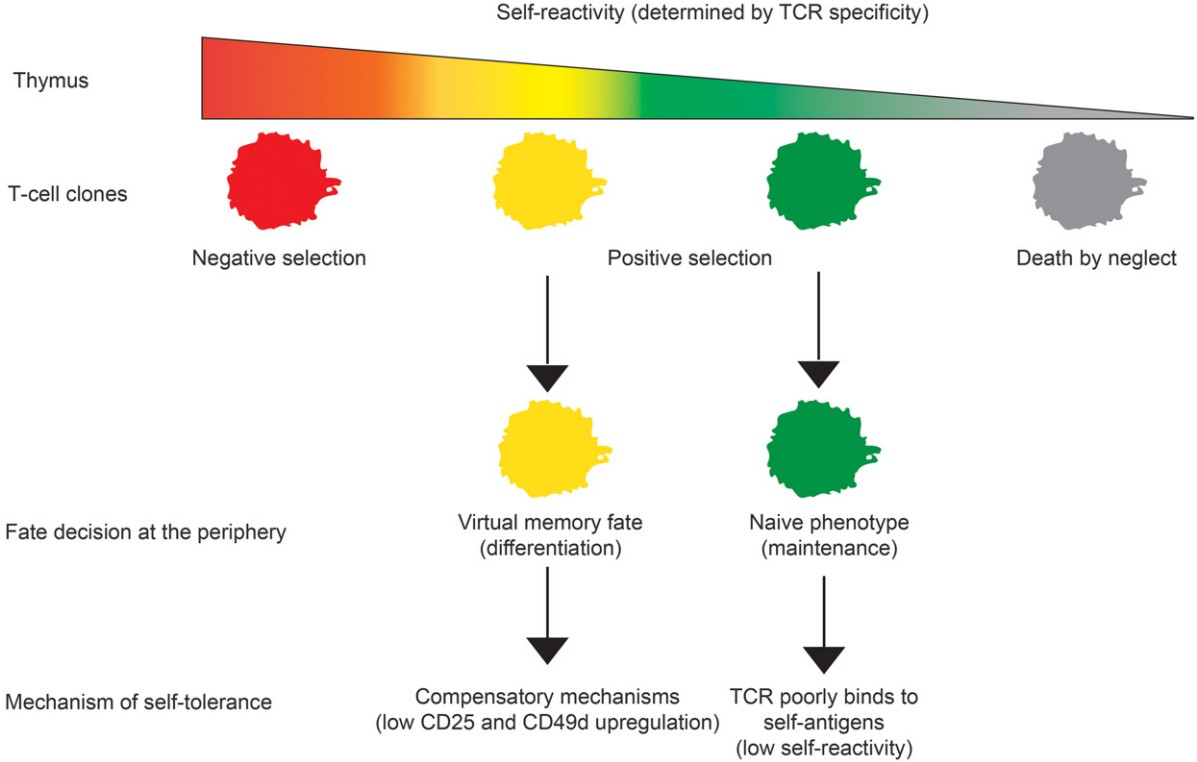

**Figure 7.  Schematic representation of the role of self-reactivity in major cell fate decisions of conventional CD8⁺ T cells.**

Our results establish a novel T-cell fate decision checkpoint, differentiation of positively selected T-cell clone with a relatively high level of self-reactivity into virtual memory T cells.

*et al*, 2017). The main argument supporting this hypothesis is that germ-free mice, with low levels of foreign antigenic exposure, contain comparable levels of CM-phenotype T cells as control mice (Haluszczak *et al*, 2009). However, in mice with normal microbiota, which were used for the subsequent characterization of VM T cells, it is difficult to exclude the existence of cross-reactive memory T cells that were previously exposed to another foreign antigen (Su *et al*, 2013). Importantly, we show that CD8.4 knock-in mice with enhanced homeostatic TCR signaling exhibit larger VM compartment than CD8WT T cells in SPF and germ-free conditions. We confirmed that VM T cells have lower expression of CD49d than antigen-experienced cells using germ-free mice and confirmed that

antigen-inexperienced VM T cells can be defined as $CD49^-CD122^{HI}$ T cells. Altogether these data established that the strength of homeostatic signals provided to T cells is a major factor leading to formation of VM compartment independently of stimulation with cognate foreign antigens.

It has been observed that IL-15 availability is a limiting factor regulating the size of the VM subset (Sosinowski *et al*, 2013; White *et al*, 2016). In this study, we showed that the intrinsic sensitivity of the TCR signaling machinery (specifically the CD8-Lck coupling frequency) is another major factor that sets the frequency of VM $CD8^+$ T cells in the secondary lymphoid organs. It has been suggested that the level of CD5, a marker of self-reactivity, is linked with the T-cell ability to form VM T cells (White *et al*, 2016). We investigated individual T-cell clones using transgenic cells with normal and hypersensitive TCR signaling machinery, comparing TCR repertoires of naïve and VM T cells, and analyzing retrogenic monoclonal T-cell populations. These complementary approaches revealed that VM T-cell formation absolutely depends on the level of self-reactivity of a particular T cell and exhibits a threshold behavior. Relatively highly self-reactive T cell clones frequently differentiate into VM T cells (~ 40–50%), whereas weakly self-reactive T cells completely lack this property. This finding characterizes the formation of VM T cells as a previously unappreciated T-cell fate decision check point, where the intensity of homeostatic TCR signals is the critical decisive factor. Our data also explain a previous observation that VM T cells are formed exclusively from T cells expressing endogenous recombined TCR chains in OT-I $Rag^+$ mice during aging (Renkema *et al*, 2014). Some of the T-cell clones that replaced the OT-I TCR with a variable endogenous one are probably more self-reactive than OT-I T cells, which drives their differentiation in VM T cells.

The functionality of a T-cell subset is determined by its gene expression profile. Whereas it is clear that VM T cells substantially differ from naïve T cells (Haluszczak *et al*, 2009; Lee *et al*, 2013; Sosinowski *et al*, 2013; White *et al*, 2016), the CD49d and CD122 were the only markers that can distinguish VM T cells from true memory (TM) T cells. In this study, we characterized gene expression of VM T cells and compared it to naïve T cells from germ-free mice and foreign antigen-specific TM T cells. Analysis focusing on previously established naïve and memory T-cell signature genes revealed that VM T cells have an intermediate gene expression profile between naïve and TM T cells. Accordingly, expression of chemokines and cytokines was generally lower in VM T cells than in TM T cells. These data suggest that VM T cells trigger a partial memory program. Alternatively, TM $CD8^+$ T cells might represent a heterogeneous population of two or more subsets with different degrees of similarity to VM T cells, as suggested by heterogeneous expression of CX3CR1 and NRP1, two genes that showed a large difference between TM and VM T cells. Indeed, CX3CR1 has been proposed as a marker that discriminates different subsets of memory T cells (Bottcher *et al*, 2015; Gerlach *et al*, 2016). Single-cell gene expression profiling would show whether immune responses to foreign antigens generate any TM T cells identical to VM T cells.

VM T cells share some phenotypic traits with stem-like memory (SCM) T cells, including higher expression of CD122, CXCR3, and dependency on IL-15 (Zhang *et al*, 2005). However, unlike VM cells, SCM T cells are derived from $CD44^{low}$ population and human SCM T-cell counterparts are derived from the $CD45RA^+$ population (Gattinoni *et al*, 2011). Furthermore, SCM T cells express Sca-1 stemness marker which is not upregulated in the VM T cells as revealed by our RNAseq data. Thus, VM and SCM T cells represent two distinguishable subsets of $CD8^+$ T cells. However, because SCM T cells give rise to central memory, effector memory, and effector $CD8^+$ T cells (Gattinoni *et al*, 2011), we cannot exclude that VM T-cell population preferentially arise from SCM T cells.

VM T cells were shown to surpass naïve T cells in their response to inflammatory cytokines IL-12 and IL-18 (Haluszczak *et al*, 2009), in the rapid generation of short-lived effectors (Lee *et al*, 2013), rapid IFNγ production, and in the protection against Lm both in antigen-specific (Lee *et al*, 2013) and in by-stander manners (Wu *et al*, 2010). We demonstrated that VM T cells develop from relatively strongly self-reactive T-cell clone. Both the hyperresponsivness and self-reactivity of VM T cells might potentially enhance their capacity to break self-tolerance. We addressed the potency of VM T cells to induce experimental autoimmune pathology by using two monoclonal models for comparing naïve and VM T cells with the same TCR specificity. We observed that VM T cells were not more efficient than naïve T cells in inducing the experimental autoimmune diabetes on a per cell basis. This can be at least partially explained by the fact that VM T cells show lower upregulation of CD25 and VLA-4 than naïve T cells when activated with a suboptimal antigen. VLA-4 has been previously shown to be essential for the induction of the tissue pathology in the mouse experimental model of type I diabetes (King *et al*, 2012). We propose that VM T cells surpass naïve T cells in some kind of responses to promote rapid immunity to pathogens (Lee *et al*, 2013; White *et al*, 2016), but they also develop compensatory mechanisms that make these cells self-tolerant to an extent comparable to naïve T cells. We used an experimental model of autoimmune diabetes, a prototypic autoimmune pathology that involves self-reactive $CD8^+$ T cells. The important aspect of our model is that the adoptively transferred neoself-reactive T cells developed in the absence of the neoself antigen. We cannot exclude the possibility that VM T cells represent a major risk in other types of autoimmune diseases/conditions.

Based on pilot studies in the field (Pihlgren *et al*, 1996; Curtsinger *et al*, 1998; London *et al*, 2000), it was generally accepted that one feature of immunological memory is that a memory T cell elicits a faster and stronger response to cognate antigens than a naïve T cell. However, recent evidence showed that, at least under certain conditions, the response of naïve T cells to an antigen is stronger than the response of memory T cells (Knudson *et al*, 2013; Mehlhop-Williams & Bevan, 2014; Cho *et al*, 2016). We showed that true memory T cells surpass virtual memory T cells in the upregulation of KLRG1, CD25, and in their potency to induce experimental autoimmune pathology. These data correspond to a previous study showing stronger responses of TM T cells in comparison with lymphopenia-induced memory T cells (Cheung *et al*, 2009), suggesting that virtual memory and lymphopenia-induced memory T cells might have similar functions. The physiological role of VM T cells needs to be further investigated, but it seems plausible that VM T cells have unique type of responses to pathogens and thus contribute to functional diversity of T-cell immunity, which might be required for efficient immune protection.

Recently, it has been proposed that human innate Eomes[+] KIR/NKG2A[+] CD8[+] T cells (Jacomet et al, 2015) represent counterparts of murine VM T cells, although expression of some markers including CD27 and CD5 substantially differed between these two subsets (White et al, 2016). It would be interesting to elucidate whether the human Eomes[+] KIR/NKG2A[+] CD8[+] subset shows similar gene expression pattern as mouse VM T cells, whether they originate from relatively highly self-reactive clones, and whether these cells acquire tolerance mechanisms as murine VM CD8[+] T cells do.

# Materials Methods

### Antibodies and reagents

Antibodies to following antigens were used for flow cytometry: CD69 (clone H1.2F3), CD11a (LFA-1) (clone M17/4), CD25 (PC61), CD3 (145-2C11), IFNγ (XMG1.2), TCRβ (H57-597), TCR Vα2 (B20.1), TCR Vα8.3 (B21.14), TCR Vα3.2 (R3-16), CD49d (R1-2), CD5 (53-7.3) (all BD Biosciences), Tbet (4B10), Eomes (Dan11mag), CD8α (53-6.7), CD8β (H35-17.2), CD127 (A7R34) (all eBioscience) CD44 (IM7), CD4 (RM-45), CD62L (MEL-14), CD122 (TM-beta1), KLRG1 (2F1), PD-1 (RMP1-30), CD19 (6D5) (all Biolegend), pErk1/2 (D13.14.4E, Cell Signaling). The antibodies were conjugated with various fluorescent dyes or with biotin by manufacturers. K$^b$-OVA PE tetramer was prepared as described previously (Stepanek et al, 2014). Peptides OVA$_{257-264}$ (SIINFEKL), Q4R7 (SIIRFERL), Q4H7 (SIIRFEHL), NP68 (ASNENMDAM), NP372E (ASNENMEAM), Mapk8$_{267-274}$ (AGYSFEKL), and Catnb$_{329-336}$ (RTYTYEKL) were purchased from Eurogentec or Peptides&Elephants. Proliferation dye CellTrace Violet was purchased from ThermoFisher Scientific (C34557).

### Flow cytometry and cell counting

For the surface staining, cells were incubated with diluted antibodies in PBS/0.5% gelatin or PBS/2% goat serum on ice. LIVE/DEAD near-IR dye (Life Technologies) or Hoechst 33258 (Life Technologies) was used for discrimination of live and dead cells. For the intracellular staining, cells were fixed and permeabilized using Foxp3/Transcription Factor Staining Buffer Set (eBioscience, 00-5523-00). For some experiments, enrichment of CD8[+] T cells was performed using magnetic bead separation kits EasySep (STEMCELL Technologies) or Dynabeads (Thermo Fisher Scientific) according to manufacturer's instructions prior to the analysis or sorting by flow cytometry. Cells were counted using Z2 Coulter Counter (Beckman) or using AccuCheck counting beads (Thermo Fisher Scientific) and a flow cytometer. Flow cytometry was carried out with a FACSCantoII, LSRII, or a LSRFortessa (BD Bioscience). Cell sorting was performed using a FACSAria III or Influx (BD Bioscience). Data were analyzed using FlowJo software (TreeStar).

### Experimental animals

All mice were 5–12 weeks old and had C57Bl/6j background. RIP.OVA (Kurts et al, 1998), OT-I Rag2$^{-/-}$ (Palmer et al, 2016), CD8.4, F5 Rag1$^{-/-}$ (Erman et al, 2006), and Vβ5 (Fink et al, 1992)

strains were described previously. Mice were bred in our facilities (SPF mice: University Hospital Basel and Institute of Molecular Genetics; germ-free mice: University of Bern, Switzerland) in accordance with Cantonal and Federal laws of Switzerland and the Czech Republic. Animal protocols were approved by the Cantonal Veterinary Office of Basel-Stadt, Switzerland, and Czech Academy of Sciences, Czech Republic. Transfer into germ-free conditions was performed using time-mating followed by transferring 2-cell embryos into germ-free foster mothers.

Males and females were used for the experiments. Age- and sex-matched pairs of animals were used in the experimental groups. If possible, littermates were equally divided into the experimental groups. The randomization for adoptive transfer experiments was done by assigning the experimental conditions to recipient mouse ID numbers by an experimenter who had no prior contact with the mice. Other experiments were not randomized. The experiments were not blinded since no subjective scoring method was used.

### RNA sequencing

RNA was isolated using Trizol (Thermo Fisher Scientific) followed by in-column DNase treatment using RNA clean & concentrator kit (Zymo Research). The library preparation and RNA sequencing by HiSeq2500 (HiSeq SBS Kit v4, Illumina) were performed by the Genomic Facility of D-BSSE ETH Zurich in Basel. Obtained single-end RNAseq reads were mapped to the mouse genome assembly, version mm9, with RNA-STAR (Dobin et al, 2013), with default parameters except for allowing only unique hits to genome (outFilterMultimapNmax = 1) and filtering reads without evidence in spliced junction table (outFilterType = "BySJout"). All subsequent gene expression data analysis was done within the R software (R Foundation for Statistical Computing, Vienna, Austria). Raw reads and mapping quality were assessed by the qQCreport function from the R/Bioconductor software package QuasR (version 1.12.0; Gaidatzis et al, 2015). Using RefSeq mRNA coordinates from UCSC (genome.ucsc.edu, downloaded in July 2013) and the qCount function from QuasR package, we quantified gene expression as the number of reads that started within any annotated exon of a gene. The differentially expressed genes were identified using the edgeR package (version 3.14.0; Robinson et al, 2010). We generated lists of naïve and memory signature genes based on previously published studies [gene sets M3022, M5832, M3039 for memory, and M3020, M5831, M3038 for naïve T cells in the Molecular Signature Databases (Kaech et al, 2002; Subramanian et al, 2005; Luckey et al, 2006; Wherry et al, 2007)]. In our memory and naïve signature gene lists, we included only genes that were listed at least in two out of the above-mentioned three respective gene sets. For the global comparison of the expression of the signature genes in naïve, VM, and true CM T cells, we used self-contained gene set enrichment test called Roast, which is available in edgeR package (Wu et al, 2010).

### DNA cloning and production and viruses

RNA was isolated using Trizol reagent (Thermo Fisher Scientific) and RNA clean & concentrator kit (Zymoresearch, R1013). Reverse transcription was performed using RevertAid (Thermo Fisher Scientific) according to the manufacturer's instructions. TCR sequences

were amplified using cDNA from sorted T cells by PCR using Phusion polymerase (New England Biolabs) and following primers: TRACrev (EcoRI) 5′-TCAGACgaattcTCAACTGGACCACAGCCTCA, TRAV14for (XhoI) 5′ GTAGCTctcgagATGGACAAGATCCTGACA GCA, TRAV12for (XhoI) 5′ GTAGCTctcgagATGCGTCCTGDCACCTG CTC and ligated into pBlueScript vector using T4 ligase (New England Biolabs) and sequenced by Sanger sequencing using T7 primer 5′ TAATACGACTCACTATAGGG. Selected clones were cloned into MSCV-GFP vector via EcoRI and XhoI (New England Biolabs).

Coding sequence of SCF, IL-3 and IL-6 was obtained from bone marrow cDNA with Phusion polymerase using these primers: SCFfor 5′-TTGGATCCGCCACCATGAAGAAGACACAAACTTGGATT ATC, SCFrev 5′-AACTCGAGTTACACCTCTTGAAATTCTCTCTCTTTC, IL-3for 5′-TTGAATTCGCCACCATGGTTCTTGCCAGCTCTACCACCAG, IL3rev 5′-AACTCGAGTTAACATTCCACGGTTCCACGGTTAGG, IL-6 for 5′-TTGAATTCGCCACCATGAAGTTCCTCTCTGCAAGAGACTT, IL6 rev 5′ AACTCGAGCTAGGTTTGCCGAGTAGATCTCAAAGTG. cDNA was cloned into pXJ41 expression vector using BamHI or EcoRI and XhoI restriction sites and sequenced. Cytokines were produced in HEK293 cells transfected with pXJ41 using polyethylenimine (PEI) transfection in ratio 2.5 μl PEI to 1 μg DNA). Supernatant was harvested 3 days after transfection. Titration against commercial recombinant cytokines of known concentration and their effect on BM cell proliferation *in vitro* was used to determine biological activity of cytokines in supernatant. Dilution of supernatant corresponding to concentration of 100 ng/ml SCF, 20 ng/ml IL-3 and 10 ng/ml IL-6 was used for cultivation of immortalized bone marrow cells.

Retroviral MSCV and pMYs particles were generated by transfection of the vectors into Platinum-E cells (Cell Biolabs) by PEI as described above.

### Ex vivo activation assay

For the analysis of IFNγ production, T cells ($1 \times 10^6$/ml in RPMI/ 10% FCS) were stimulated with 10 ng/ml PMA and 1.5 μM ionomycin or 1 μM OVA peptide in the presence of BD Golgi Stop for 5 h. For the CD69 and CD25 upregulation assay, dendritic cells differentiated from fresh or immortalized bone marrow stem cells were pulsed with indicated concentration of indicated peptides, mixed with T cells isolated from LNs in a 1:2 ratio, and analyzed after ~ 16 h of coculture as described previously (Palmer *et al*, 2016).

### Bone marrow chimeras

Bone marrow cells were isolated from long bones of indicated mouse strains and lymphocytes were depleted using biotinylated antibodies to CD3 and CD19 and Dynabeads biotin binder kit (Thermo Fisher Scientific). In total, $6 \times 10^6$ cells (always a 1:1 mixture from two different donor strains) in 200 μl of PBS were injected into irradiated (300 cGy) Rag2$^{-/-}$ host mice i.v. The mice were analyzed 8 weeks after the transfer.

### Monoclonal retrogenic T cells

Generation of immortalized bone marrow was described previously (Ruedl *et al*, 2008). Briefly, Vβ5 Rag2$^{-/-}$ mice were treated with

100 mg/kg 5-fluorouracil and bone marrow cells were harvested 5 days later. Cells were cultivated in complete IMDM (10% FCS) supplemented with SCF, IL-3, and IL-6. After 2 days, proliferating cells were virally transduced with a fusion construct NUP98-HOXB4 in a retroviral vector pMYs. Viral infections were performed in the presence of 10 μg/ml polybrene by centrifugation (90 min, 1,250 *g*, 30°C). The transduced cells were selected for 2 days in puromycin (1 μg/ml). Selected immortalized cells were subsequently virally transduced with MSCV vector containing s TCRα-encoding gene and GFP as a selection marker. Two days after the transduction, GFP$^+$ was FACS-sorted and transplanted into irradiated (7 Gy) congenic Ly5.1 recipients. At least 8 weeks after the transplantation, the recipient mice were sacrificed and donor LN T cells were used for cell fate analysis by flow cytometry or for adoptive transfers.

### In vivo activation and a model for autoimmune diabetes

Indicated numbers of T cells were adoptively transferred into a host mouse i.v. On a following day, the host mice were immunized with indicated peptide (50 μg) and LPS (25 μg) in 200 μl of PBS i.p. or with 5,000 CFU of Lm. Lm strains expressing OVA, Q4R7, and Q4H7 have been described previously (King *et al*, 2012; Oberle *et al*, 2016). Lm expressing NP68 was produced by adding the ASNENMDAM epitope to ovalbumin encoding gene and introduced to Lm as previously described (Zehn *et al*, 2009). Dendritic cells for *in vivo* experiments were generated from full bone marrow isolated from long bones of 6- to 10-week-old mice. Cells were cultured for 10 days in complete Iscove's modified Dulbecco's medium (10% FCS) conditioned with 2% GM-CSF supernatant (Lutz cells). Medium was refreshed for new on day 4 and 7. Differentiated DCs were pulsed with corresponding peptide ($10^{-7}$ M) in the presence of LPS (100 ng/ml) for 3 h and $1 \times 10^6$ antigen-loaded DCs were used for i.v. immunization. In the experimental model of autoimmunity, we monitored glucose in the urine of RIP.OVA mice on a daily basis using test strips (Diabur-Test 5000, Roche or GLUKO-PHAN, Erba Lachema, Czech Republic). The animal was considered to suffer from lethal autoimmunity when the concentration of glucose in the urine reached ≥ 1,000 mg/dl. We also measured blood glucose by contour blood glucose meter (Bayer) on day 7 post-infection. In the diabetic experiments, mice that died before the end of the monitored period (14 days) and before they reached 1,000 mg/dl glucose levels in the urine were excluded. Only one mouse in total was excluded based on this pre-established criterium.

### Generation of true memory T cells

True memory T cells were generated by infecting Vβ5 mice with 5,000 CFU of Lm-OVA. After 60–90 days (for RNAseq) or 30–50 days (for FACS staining), CD8$^+$ Kb-OVA tetramer$^+$ CD49d$^+$ CD44$^+$ CD62L$^+$ from LNs and the spleen were sorted (or gated). TM OT-I T cells were generated by adoptive transfer of $1 \times 10^6$ T cells from OT-I Rag2$^{-/-}$ mouse into Ly5.1 recipient and subsequent infection with 5,000 CFU of Lm-OVA. At least 30 days after infection, CD8$^+$ Ly5.2$^+$ cells from LNs and the spleen were sorted and adoptively transferred into recipient mice.

### *Listeria* clearance

FACS-sorted $1 \times 10^4$ naïve CD8WT OT-I, CD8.4 OT-I (VM), sorted true memory OT-I T cells, or no T cells were adoptively transferred into Ly5.1 C57Bl/6j mice followed by infection with 5,000 CFU Lm-Q4H7. The recipient mice were sacrificed on day 3 and 5 post-infection, and the spleen was homogenized and lysed in PBS with 0.1% Tergitol (Sigma-Aldrich). 1/20 of splenic lysate was plated onto brain–heart infusion agar (BHI, Sigma-Aldrich) plates with 200 μg/ml streptomycin and incubated at 37°C. The resulting number of colonies was quantified the following day.

### Statistics

Statistical analysis was carried out using Prism (V5.04, GraphPad Software) or MS Excel. The statistical tests are indicated for each experiment. Whenever possible, we used nonparametric statistical tests. In one case, we use one-value *t*-test after the data passed Shapiro–Wilk normality test. In the *ex vivo* activation assays, we used paired Student's *t*-test (pairs = individual experiments). In this case, we tested the normality of differences using Shapiro–Wilk test, using pooled differences from two highest concentrations of peptides for each condition (because of too few data points for each peptide concentration). For comparing the abundance of individual TCR clones in different subsets, we use paired *t*-test as a post-test after using global chi-square test. Because of the very low $n = 3$, we could not test the normality of differences. All statistical tests were two-tailed.

### Data availability

The data RNAseq data are deposited in the GEO database (GSE90522).

**Expanded View** for this article is available online.

### Acknowledgements

We thank Prof. Alfred Signer for providing us with CD8.4 mice. We thank Ladislav Cupak, Barbara Hausmann, and Rosmarie Lang for their technical assistance and genotyping of mice. We thank Zdenek Cimburek and Matyas Sima for cell sorting. We thank Prof. Ed Palmer for his multisided support to the project. This project was supported by the Swiss National Science Foundation (Promys, IZ11Z0_166538), the Czech Science Foundation (GJ16-09208Y) to OS, ERC Starting Grant (ProtecTC) to DZ, and the Institute of Molecular Genetics (RVO 68378050). The animal facility of the IMG is a part of the Czech Centre for Phenogenomics and the work there was supported in part by following grants: LM2015040, OP RDI CZ.1.05/2.1.00/19.0395, OP RDI BIOCEV CZ.1.05/1.1.00/02.0109 provided by the Czech Ministry of Education, Youth, and Sports and the European Regional Development Fund. The Group of Adaptive Immunity at the Institute of Molecular Genetics in Prague is supported by an EMBO Installation grant. AM, VH, and MP are students of the Faculty of Science, Charles University, Prague.

### Author contributions

AD, AM, DM, MH, and OS planned, performed, and analyzed experiments. VH and MP performed and analyzed experiments. PD analyzed experiments. RI did the bioinformatic analysis of the RNA sequencing data. SO and DZ prepared transgenic Lm strains and provided Vβ5 mice. KDM established and managed the germ-free strains. OS conceived the study. OS, AD, AM, and PD wrote the manuscript. All authors commented on the manuscript.

### Conflict of interest

The authors declare that they have no conflict of interest.

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
