## [Review Process File · The EMBO Journal]

Strong homeostatic TCR signals induce formation of self-tolerant virtual memory CD8 T cells

Ales Drobek, Alena Moudra, Daniel Mueller, Martina Huranova, Veronika Horkova, Michaela Pribikova, Robert Ivanek, Susanne Oberle, Dietmar Zehn, Kathy D. McCoy, Peter Draber and Ondrej Stepanek.

Review timeline:

Submission date:	28 th October 2017
Editorial Decision:	11 th December 2017
Revision received:	11 th March 2018
Accepted:	9 th April 2018

Editor: Karin Dumstrei

Transaction Report:

1st Editorial Decision

11th December 2017

Thank you for submitting your manuscript to The EMBO Journal. Your study has now been seen by three referees and their comments are provided below.

As you can see from the comments, the referees find the analysis interesting. However they also raise some important points that have to be resolved. Should you be able to address the concerns raised in full then I am happy to consider a revised version. Let me know if we need to discuss anything further.

I should add that it is EMBO Journal policy to allow only a single major round of revision and that it is therefor important to resolve the major concerns raised at this stage.

When preparing your letter of response to the referees' comments, please bear in mind that this will form part of the Review Process File, and will therefore be available online to the community. For more details on our Transparent Editorial Process, please visit our website: http://emboj.embopress.org/about#Transparent_Process

Thank you for the opportunity to consider your work for publication. I look forward to your revision.

REFeree REPORTS

Referee #1:

The authors study the development and function of "virtual memory" CD8⁺ T cells, a population antigen-inexperienced T lymphocytes that nevertheless have properties of memory cells, and are believed to arise by homeostatic mechanisms. While TCR/coreceptor signal strength has been proposed as a major factor in driving development of virtual memory cells, the authors propose direct data in support of this idea, through use of mice engineered to have a CD8 molecule that associates more strongly with the kinase Lck. The authors go on to extend current understanding of how virtual and "true" memory T cells differ in gene expression and function, reaching the conclusion that virtual memory cells are a distinct population, yet retain many characteristics of memory cells. Interestingly, however, virtual memory cells were no more potent than naïve cells of the same specificity in driving autoimmune disease, in a diabetes model.

These studies are interesting and present novel and unexpected findings about how the virtual memory population arises and is transcriptionally and functionally distinct from "true" antigen-driven memory cells. There are a number of concerns, however.

1) In Fig. 6, the authors make the somewhat surprising finding that the virtual memory cells (derived from CD8.4 OT-I mice) are no better than naïve CD8⁺ T cells (from normal OT-I mice) are driving diabetes in RIP-OVA mice, following priming with low (or high) affinity peptide/MHC ligands. There are a few concerns with this experiment however. First, it is not clear whether the authors are in a dynamic range where differences between naïve and memory CD8⁺ T cells would be expected - there is no control using "true" memory OT-I to show whether THESE cells WOULD exhibit faster, more penetrant autoimmunity than naïve OT-I, under these conditions. Hence, it is not clear whether the failure to see a difference in these assays is unique to virtual memory cells, or would equally well apply to true memory cells of the same specificity. This should be tested directly.

2) Second, along the same lines, it is unclear whether the lack of increased autoimmunity by the CD8.4 OT-I virtual memory cells is a consequence of faster/more complete eradication of the *Listeria* recombinant bacteria used to prime the response - it is possible that priming, especially with the strain carrying the low affinity Q4H7 variant is less effective when virtual memory OT-I are used because the *Listeria* elimination is MORE effective. This could certainly be quantified (bacterial load on successive days), and other ways to induce the response (e.g peptide immunization, with suitable adjuvants) should be explored to test this idea.

3) The TCR repertoire data in Fig. 3 are intriguing - suggesting strong bias of certain OVA/Kb specific clones to produce virtual memory cells. Since these "VM clones" produce very high numbers of virtual memory cells without the assistance of CD8.4 (as used for OT-I), it would be intriguing to know whether their capacity to induce diabetes (in the model used for Fig. 6) would be strong than a comparable "naïve" clone (comparable meaning, for example, if the OVA/Kb tetramer staining was equivalent). Part of the reason for investigating this is that there may be unexpected consequences in activation of cells bearing CD8.4, so it would be valuable to be able to compare reactivity of VM and naïve clones bearing the physiological CD8 molecule. Have the authors conducted such studies?

Referee #2:

The manuscript, "Strong homeostatic TCR signals induce formation of self-tolerance virtual memory CD8 T cells" by Drobeck, et al. examines the mechanistic origins of foreign antigen-naïve memory T cells. The origins and consequences of these "virtual memory" (VM) cells, which express markers associated with CD8⁺ T cell memory (CD44⁺) in the absence of prior stimulation by foreign antigen, is of considerable interest as they could conceivably affect the T cell repertoire available to mediate protective immune responses. Furthermore, given previous demonstrations that VM cell development is dependent upon TCR recognition of self-peptide:MHC ligands, VM cells could potentially pose an autoimmune risk.

The authors hypothesize that strong TCR signaling in response to self-antigens induces a fate-

determining decision to develop into VM T cells. The authors present well-designed and convincing experiments to demonstrate that VM cell generation is a selective process driven by proximal TCR signaling. Using the chimeric 8.4 co-receptor, which promotes increased recruitment of Lck to the immunological synapse, they demonstrate a dose-dependent effect of TCR signaling specifically for TCRs with higher affinity for self-ligands in driving formation of VM cells. Performing these experiments in germ-free mice, the authors add a level of certainty to the "antigen inexperienced" nature of VM cells that has not been previously demonstrated. Comparisons of the TCR repertoires of VM and non-VM cells recognizing the same foreign antigen:MHC ligand, and subsequent functional investigation using retroviral expression, provide further novel and convincing evidence that VM cell generation is an instructive process dependent on TCR ligand recognition and signaling.

Comparative gene expression analysis of VM, naïve, and memory cells provide reasonable evidence that VM cells represent a distinct state from either naïve or antigen-experienced memory cells. Hierarchical analysis is supportive of a model that places VM cells as an intermediate state between naïve and antigen-experienced true memory cells. These data provide important and convincing evidence of VM cells existing as a true "subset" in T cell development.

The data demonstrating the influence of TCR signal strength, the increased assurance of antigen-inexperienced nature through the use of germ-free mice, and the gene expression data demonstrating VM cells as a distinct "intermediary" developmental stage are novel and provide significant new insight into the development of VM cells. However, the functional studies are problematic and uninformative (concerns described below), which significantly limit the impact of the report and draw possibly incorrect conclusions in regard to the autoimmune potential of VM cells. These problems preclude publication of the report in its current form.

Minor Concerns

The gene expression data support a model where VM cells are a true separate "intermediary state" between naïve and antigen-experienced true memory cells. However, the more interesting question would be the transitional relationship between the subsets- are VM cells predisposed to become memory cells or short-term effector cells upon response to foreign antigen? Differences in the response to antigenic stimulation as compared to naïve cells could have important implications regarding the biologic importance of VM cells.

FYI--In addition to the generation of VM cells in germ-free mice, Surh and co-workers showed that these cells are generated in completely antigen-free mice (Kim et al., *Science* 19 FEBRUARY 2016 • VOL 351 ISSUE 6275: Supplementary Figure 3). In addition, there are parallels between the VM T cells and HSP T cells that have been previously compared with bona fide memory cells: Cheung et al., *JCI* 118:3362 (2009).

19 FEBRUARY 2016 • VOL 351 ISSUE 6275

Major Concerns

The experiments presented in the last figure may not support the overall conclusions of the paper. First, what is the question? Is it that VM cells have a high propensity for autoimmunity by virtue of their higher affinity for antigen/MHC? Or is it that T cells that have made the transition to the VM state acquire a hyperactivity that is more prone to autoimmunity. More likely, given the data presented here, these two concepts cannot be untangled. It seems to this reviewer that the real question is whether in an unaltered, SPF mouse, do the autoreactive cells that can be induced by various means come preferentially from the starting VM subset? With the identification of higher affinity TCRs in the Vb-transgenic mice, the authors would be a position to test this.

The authors state, "Because CD8.4 T cells have stronger reactivity to antigens than CD8WT T cells, this monoclonal T-cell model corresponds to the physiological situation where TCRs of VM T cells are intrinsically more reactive to self-antigens than TCRs of naïve T cells." I find this to be somewhat circular logic. The high-reactive CD8.4 cells produce more VM cells, but comparing these cells to naïve cells is not informative with respect to VM vs. naïve T cells. There are two variables and one comparison: hyperactive + VM phenotype vs. naïve cells. Such an experiment might be carried out by sorting VM cells and naïve cells from OT-I mice (and OT-I;CD8.4 mice), and comparing these two populations for their ability to cause disease.

In addition, is this really an autoimmune model? The OT-I T cells did not develop in a host that contained OVA or Q4H7. What they are really testing is whether Lm-OVA or Lm-L4H7 can prime transferred OT-I or OT-I;CD8.4 cells to enter the islets and kill OVA-expressing b-islet cells. For these T cells, OVA is a foreign antigen, not the low-affinity ligand driving positive selection and HSP (and presumably VM cell formation). The positive selecting ligand is known in the OT-I system (Hogquist KA. 1997. *Immunity*. 6:389) and examination of reactivity by naïve and VM cells

against this ligand might be more informative. In the model used in this investigation, OVA simply represents a cognate foreign antigen to the transferred T cells, regardless of naïve or VM status. Although this manuscript describes interesting and important experiments with respect to virtual memory cells, I do not believe that it addresses the role of these cells in autoimmune processes.

Referee #3:

The authors describe factors that determine the generation of virtual memory T cells and conclude that despite their self-reactivity and partial differentiation they do not seem more prone to trigger autoimmunity than naïve T cells. This study addresses the detailed characterization of memory-phenotype CD8 T cells and comes as a significant contribution to the body of literature accumulated since at least 12 years. What is really interesting about this manuscript is the high granularity of the molecular and functional characterization of virtual memory (VM) in comparison to naïve and true memory T cells, using cutting edge tools and animal models. As the authors point out, the novelty of the present study stems from their showing how the T cell intrinsic sensitivity to TCR-originated signals determines the magnitude of naïve T cell transit to virtual memory T cells. Moreover, the RNA-seq analyses convincingly place the VM subset as an intermediate between naïve and central memory T cells. In short this report may significantly enrich the description of the phenotypical makeup and functional competence of memory-phenotype CD8 T cells.

This study is well performed and clearly written. A minor concern is the repetitious nature of the discussion section relative to the content of the results. It may strengthen the manuscript to avoid too much repetition in the discussion. In addition, the authors may discuss how these high resolution-defined VM T cells relate to the so called stem like memory T cells. The recent reports on both mouse and human stem cell like memory T cells allow to place this subset between naïve and memory T cells. Some of the phenotypic traits for the latter cells are shared with those of VM T cells. An so does the overall functional competence they have displayed. It would be interesting for the readers following the characterization of T cell memory to know how the authors may bridge this "new" subset with the VM T cell subset. There is a need for integration and streamlining of the memory cell subsets.

Minor points:

Page 3, paragraph 3, numeral (ii) is duplicated.

Figure 3G: the nomenclature of the naïve TCR retrogenics differs slightly from that provided in the text (page 7, next to last paragraph).

Referee #1:

The authors study the development and function of "virtual memory" CD8+ T cells, a population of antigen-inexperienced T lymphocytes that nevertheless have properties of memory cells, and are believed to arise by homeostatic mechanisms. While TCR/coreceptor signal strength has been proposed as a major factor in driving development of virtual memory cells, the authors propose direct data in support of this idea, through use of mice engineered to have a CD8 molecule that associates more strongly with the kinase Lck. The authors go on to extend current understanding of how virtual and "true" memory T cells differ in gene expression and function, reaching the conclusion that virtual memory cells are a distinct population, yet retain many characteristics of memory cells. Interestingly, however, virtual memory cells were no more potent than naïve cells of the same specificity in driving autoimmune disease, in a diabetes model.

These studies are interesting and present novel and unexpected findings about how the virtual memory population arises and is transcriptionally and functionally distinct from "true" antigen-driven memory cells. There are a number of concerns, however.

We are very thankful for the positive evaluation of our results and for the very useful comments. We addressed these comments in the revised manuscript (see below).

1) In Fig. 6, the authors make the somewhat surprising finding that the virtual memory cells (derived from CD8.4 OT-I mice) are no better than naïve CD8+ T cells (from normal OT-I mice) at driving diabetes in RIP-OVA mice, following priming with low (or high) affinity peptide/MHC ligands. There are a few concerns with this experiment however. First, it is not clear whether the authors are in a dynamic range where differences between naïve and memory CD8+ T cells would be expected - there is no control using "true" memory OT-I to show whether THESE cells WOULD exhibit faster, more penetrant autoimmunity than naïve OT-I, under these conditions. Hence, it is not clear whether the failure to see a difference in these assays is unique to virtual memory cells, or would equally well apply to true memory cells of the same specificity. This should be tested directly.

We appreciate this comment and we have addressed this important issue in the revised manuscript. We added a new set of experiments in which we transferred as few as 10,000 OT-I T cells into RIP.OVA mice followed by Lm-Q4H7 infection (Fig. 5B, Fig. EV5A). In the course of these experiments, we titrated the numbers of transferred neo-self-reactive OT-I T cells from 100,000 (all mice diabetic), through 50,000 (majority of mice diabetic) and 20,000 (~ 50% mice diabetic) to 10,000 (vast majority of mice protected). We believe that this gradual titration reveals the dynamic range and sensitivity of the assay. It is sensitive enough to capture ≤ 2 -fold difference in the number of transferred self-reactive T cells (50 k vs. 20k, 20k vs. 10k). This implies that the assay can reveal a difference of the autoimmune potential of two different cell types with a similar magnitude (i.e., ≤ 2 fold).

We agree with the reviewer that the side-by-side comparison of virtual and true memory T cells is very informative. The differences between naïve and true memory T cells have become a matter of debate and it is not clear, how true memory T cells would behave in these assays. First, we compared the responses of naïve, VM, and true memory T cells in their responses to LM-OVA. We demonstrated that true memory T cells have increased upregulation of KLRG1 and CD25 than naïve and VM T cells (Fig. 5G). Moreover, we compared true memory OT-I T cells and CD8.4 OT-I virtual memory T cells in the

model of autoimmune diabetes. True memory T cells were more potent than VM T cells in the induction of autoimmune diabetes (Fig. 5H, EV5D). This documents that the assay is sensitive enough to reveal differences between particular cell types. Importantly, these data show that true memory T cells are more efficient than virtual memory T cells in inducing autoimmunity in this model and provide additional support to our previous conclusions that virtual memory possess unique mechanisms of self-tolerance.

2) Second, along the same lines, it is unclear whether the lack of increased autoimmunity by the CD8.4 OT-I virtual memory cells is a consequence of faster/more complete eradication of the *Listeria* recombinant bacteria used to prime the response - it is possible that priming, especially with the strain carrying the low affinity Q4H7 variant is less effective when virtual memory OT-I are used because the *Listeria* elimination is MORE effective. This could certainly be quantified (bacterial load on successive days), and other ways to induce the response (e.g. peptide immunization, with suitable adjuvants) should be explored to test this idea.

*This is a very relevant question. However, the eradication of the relatively low number of *Listeria* used for the infection (5000 CFU) is fast. In the protection assays, usually much higher bacterial loads are used in order to observe effects of adoptively transferred antigen-specific T cells on *Listeria* clearance. Moreover, Q4H7 is a low-affinity antigen for OT-I and we expect that endogenous high-affinity *Listeria*-specific T cells are much more efficient in clearing *Lm*-Q4H7 than transferred OT-I T cells. However, we addressed this concern experimentally showing that the adoptive transfer of OT-I or CD8.4 OT-I or even true memory OT-I T cells does not have any impact on the clearance of *Lm*-Q4H7 on day 3 and 5 post-infection in our experimental setup (Fig. EV5B). Because *Listeria* is almost completely cleared on day 5 post-infection in all cases, we conclude that efficiency of *Listeria* clearance does not differ between the experimental groups.*

*We complemented our *Listeria* experiments with an experiment where naïve and VM T cells (coming from clones V14-C1 and V14-C2; see below) were stimulated with antigen-loaded bone marrow-derived dendritic cells (Fig. 6B). This experiment shows that primed VM T cells have lower expression of CD49d, similarly to the experiments in which *Listeria* was used (Fig. 5F). We did not use this type of stimulation for diabetic experiments, because large numbers of additional RIP.OVA mice would be required for titrating the numbers of self-reactive T cells and dendritic cells to find experimental conditions with sufficient dynamic range. Besides ethical issues (rule of reduction from 3R principle), we did not have enough RIP.OVA recipient mice to perform these experiments in a timely manner.*

3) The TCR repertoire data in Fig. 3 are intriguing - suggesting strong bias of certain OVA/Kb specific clones to produce virtual memory cells. Since these "VM clones" produce very high numbers of virtual memory cells without the assistance of CD8.4 (as used for OT-I), it would be intriguing to know whether their capacity to induce diabetes (in the model used for Fig. 6) would be strong than a comparable "naïve" clone (comparable meaning, for example, if the OVA/Kb tetramer staining was equivalent). Part of the reason for investigating this is that there may be unexpected consequences in activation of cells bearing CD8.4, so it would be valuable to be able to compare reactivity of VM and naïve clones bearing the physiological CD8 molecule. Have the authors conducted such studies?

This is a very relevant point and we have been working on this for some time. Because we observed substantial intrinsic differences in the responsiveness of particular OVA-reactive TCRs (not caused by the T-cell differentiation status), direct comparison of 'naïve' and 'VM' clones would not be informative. However, we took advantage of the fact that even the clones that produce high levels of VM T cells do develop population of naïve T cells as well. Therefore, we could compare naïve and VM cells formed from the identical VM clone (clones V14-C1, V14-C2). We sorted the same amount of naïve

and VM OVA-reactive cells, transferred them to RIP.OVA mice and infected them with Lm-OVA. Interestingly, virtual memory T cells were less efficient in inducing the diabetes than naïve T cells with the same specificity (Fig. 6A). This supports our previous conclusions from the CD8.4 OT-I model.

Referee #2:

The manuscript, "Strong homeostatic TCR signals induce formation of self-tolerance virtual memory CD8 T cells" by Drobeck, et al. examines the mechanistic origins of foreign antigen-naïve memory T cells. The origins and consequences of these "virtual memory" (VM) cells, which express markers associated with CD8+ T cell memory (CD44+) in the absence of prior stimulation by foreign antigen, is of considerable interest as they could conceivably affect the T cell repertoire available to mediate protective immune responses. Furthermore, given previous demonstrations that VM cell development is dependent upon TCR recognition of self-peptide:MHC ligands, VM cells could potentially pose an autoimmune risk.

The authors hypothesize that strong TCR signaling in response to self-antigens induces a fate-determining decision to develop into VM T cells. The authors present well-designed and convincing experiments to demonstrate that VM cell generation is a selective process driven by proximal TCR signaling. Using the chimeric 8.4 co-receptor, which promotes increased recruitment of Lck to the immunological synapse, they demonstrate a dose-dependent effect of TCR signaling specifically for TCRs with higher affinity for self-ligands in driving formation of VM cells. Performing these experiments in germ-free mice, the authors add a level of certainty to the "antigen inexperienced" nature of VM cells that has not been previously demonstrated. Comparisons of the TCR repertoires of VM and non-VM cells recognizing the same foreign antigen:MHC ligand, and subsequent functional investigation using retroviral expression, provide further novel and convincing evidence that VM cell generation is an instructive process dependent on TCR ligand recognition and signaling.

Comparative gene expression analysis of VM, naïve, and memory cells provide reasonable evidence that VM cells represent a distinct state from either naïve or antigen-experienced memory cells. Hierarchical analysis is supportive of a model that places VM cells as an intermediate state between naïve and antigen-experienced true memory cells. These data provide important and convincing evidence of VM cells existing as a true "subset" in T cell development.

The data demonstrating the influence of TCR signal strength, the increased assurance of antigen-inexperienced nature through the use of germ-free mice, and the gene expression data demonstrating VM cells as a distinct "intermediary" developmental stage are novel and provide significant new insight into the development of VM cells. However, the functional studies are problematic and uninformative (concerns described below), which significantly limit the impact of the report and draw possibly incorrect conclusions in regard to the autoimmune potential of VM cells. These problems preclude publication of the report in its current form.

We thank the Reviewer for positive evaluation of the major part of the manuscript and for very useful comments. We performed several experiments that significantly improved our understanding of in vivo roles of VM T cells and their potential to induce autoimmune pathology.

Minor Concerns

The gene expression data support a model where VM cells are a true separate "intermediary state" between naïve and antigen-experienced true memory cells. However, the more interesting question would be the transitional relationship between the subsets- are VM cells predisposed to become memory cells or short-term effector cells upon response to foreign antigen? Differences in the response to antigenic stimulation as compared to naïve cells could have important implications regarding the biologic importance of VM cells.

We are thankful for this comment. We compared the responses of naïve OT-I, CD8.4 OT-I VM, and true memory T cells to Lm-OVA. The upregulation of the effector marker KLRG1 upon stimulation with Lm-OVA showed a following hierarchy: true memory OT-I > virtual memory CD8.4 OT-I > naïve OT-I, indicating that virtual memory T cells are less efficient than true memory T cells in the upregulation of this marker of effector cells (Fig. 5G). Upregulation of CD25 is also stronger in true memory OT-I than in virtual memory CD8.4 OT-I T cells (Fig. 5G). Interestingly, CD25 was slightly more upregulated by stimulated naïve than virtual memory T cells in vivo, although the difference was not statistically significant ($p=0.06$) (Fig. 5G).

In addition, we compared the responses of naïve OT-I and CD8.4 OT-I VM to Lm-OVA and Lm-Q4H7. CD8.4 OT-I generated slightly higher percentage of short-lived effector cells (IL-7R⁺KLRG1⁺) and lower percentage of memory precursors than CD8WT OT-I upon stimulation with Lm-OVA. Although the differences in the SLEC formation were slight and nonsignificant, these data are essentially in agreement with previously published data (Lee et al. Proc Natl Acad Sci U S A. 2013 Aug 13; 110(33): 13498–13503). However, CD8.4 OT-I generated lower percentage of short-lived effector cells (IL-7R-KLRG1⁺) than CD8WT OT-I upon stimulation with Lm-Q4H7 (Fig. 5E, Fig. EV5C), suggesting that virtual memory T cells might have reduced response to suboptimal antigens. Interestingly, CD8.4 OT-I generated higher percentage of IL-7R⁺KLRG1⁺ cells than CD8WT OT-I and this difference was more pronounced in Lm-Q4H7 stimulation (Fig. 5E, Fig. EV5C). The role of IL-7R⁺KLRG1⁺ cells is unclear, but the fact that their formation associated with responses to weak antigens, they most likely do not represent cells with a strong effector potential.

FYI--In addition to the generation of VM cells in germ-free mice, Surh and co-workers showed that these cells are generated in completely antigen-free mice (Kim et al., Science 19 FEBRUARY 2016 • VOL 351 ISSUE 6275: Supplementary Figure 3). In addition, there are parallels between the VM T cells and HSP T cells that have been previously compared with bona fide memory cells: Cheung et al., JI 183:3362 (2009).

19 FEBRUARY 2016 • VOL 351 ISSUE 6275

We are thankful to the Reviewer for these references. The study by Kim et al. is very interesting, but it shows only CD4⁺ memory-like T cells in the antigen-free mice. As our study focuses exclusively on CD8⁺ VM T cells we did not cite this study in the manuscript. We mention the study by Cheung et al. in the discussion of the revised manuscript.

Major Concerns

The experiments presented in the last figure may not support the overall conclusions of the paper. First, what is the question? Is it that VM cells have a high propensity for autoimmunity by virtue of their higher affinity for antigen/MHC? Or is it that T cells that have made the transition to the VM state acquire a hyperactivity that is more prone to autoimmunity. More likely, given the data presented here, these two concepts cannot be untangled.

The Reviewer correctly names two reasons, why we initially expected that VM T cells are more potent in inducing autoimmune pathology than naïve T cells: (i) because their TCRs are more self-reactive than TCRs of naïve T cells (which is mimicked by CD8.4 in our monoclonal model) and (ii) because they have acquired the memory-like phenotype. During the revision, we carried out experiments showing that true memory T cells are indeed relatively potent in inducing the autoimmune pathology in our model system (Fig. 5H). Thus, the question addressed by the last figure (last two figures in the revised manuscript) is following: Are VM T cells superior to naïve T cells in their ability to trigger tissue pathology on per cell basis? We adjusted the manuscript to be clearer in this respect.

However, our extensive experimental work addressing this question surprisingly revealed that CD8.4 OT-I VM T cells do not show higher autoimmune potential than naïve CD8WT OT-I cells on the per cell basis in our model. Moreover, CD8.4 OT-I VM T cells exhibit some signs of hyporeactivity to (suboptimal) antigens. We would have to untangle the two scenarios (TCR reactivity and memory phenotype) only if VM T cells do exhibit higher potential to induce autoimmune pathology than naïve T cells on a per cell basis, which does not happen.

It is difficult to imagine that increased affinity of VM T cells to self-antigens (or increased Lck recruitment in the CD8.4 OT-I model) reduce their potential to induce the autoimmune tissue damage directly. Thus, we conclude that it is the VM differentiation status that reduce, rather than increase, the potential of these cells to induce the autoimmune pathology.

The Reviewer is completely right that our CD8.4 OT-I monoclonal model for virtual memory T cells has some potential caveats (CD8.4 might have some unexpected effects). For this reason, we compared the autoimmune potential of our retrogenic monoclonal naïve and VM populations expressing the same TCR (V14-C1 or V14-C2). In this setup, we compared two cell types with identical TCRs and WT CD8 (thus, the TCR's affinity to the antigen or CD8-Lck coupling is not an issue anymore). Using this model, we could solely address the role of the naïve vs. VM differentiation status in inducing the tissue pathology. Interestingly, VM had lower potential to induce the autoimmune pathology than naïve T cells expressing the same TCR (Fig. 6A), showing that VM T cells are more self-tolerant than naïve T cells in this model of experimental autoimmune pathology. These data also support our interpretation of the results from the CD8WT and CD8.4 OT-I model (Fig. 5B, 5H), i.e. that the VM differentiation stage compensates for the increased reactivity of the TCR proximal machinery induced by expression of CD8.4.

It seems to this reviewer that the real question is whether in an unaltered, SPF mouse, do the autoreactive cells that can be induced by various means come preferentially from the starting VM subset?

This is really an interesting question! We are currently developing tools for addressing whether autoreactive T cells inducing autoimmunity are coming preferentially from the naïve or VM subsets. However, we believe that this question goes beyond the scope of this manuscript. In this work, we elucidate the potential of the naïve vs. VM differentiation stages to induce autoimmune pathology on a per cell basis (i.e., do VM T cells have increased or decreased potential/capacity to induce autoimmune pathology in comparison to naïve T cells?).

Addressing the above question has a few potential issues. Even if we observe that descendants of VM or naïve T cells are overrepresented among the T-cells inducing an autoimmune pathology in an unmanipulated animal, it cannot bring a clear answer. It could be a results of two potentially counteracting effects: (i) increased affinity of these T cells to self and (ii) the naïve vs. VM differentiation status. Actually, we propose that the self-tolerant VM phenotype compensates for the

relatively high level of self-reactivity and we would expect that naïve and VM T cells might both contribute to autoimmunity in a comparable level. The key finding of this part of our study is that VM T cells are not significantly less self-tolerant than naïve T cells.

The best model for studying the role of VM formation in self-tolerance and autoimmunity would be such a model that would allow us to block the VM T cell formation from relatively highly self-reactive T cells without other disturbances to the T cell compartment. However, such a model is not available yet.

The authors state, "Because CD8.4 T cells have stronger reactivity to antigens than CD8WT T cells, this monoclonal T-cell model corresponds to the physiological situation where TCRs of VM T cells are intrinsically more reactive to self-antigens than TCRs of naïve T cells." I find this to be somewhat circular logic. The high-reactive CD8.4 cells produce more VM cells, but comparing these cells to naïve cells is not informative with respect to VM vs. naïve T cells. There are two variables and one comparison: hyperactive + VM phenotype vs. naïve cells. Such an experiment might be carried out by sorting VM cells and naïve cells from OT-I mice (and OT-I;CD8.4 mice), and comparing these two populations for their ability to cause disease.

The Reviewer is right that there are two variables in one comparison, which has both negative and beneficial aspects. The advantage of using CD8.4 OT-I and CD8WT OT-I as monoclonal models for studying VM and naïve T cells, respectively, is that it not only reflects distinct differentiation status, but also mimics the higher level of self-reactivity of VM T cells by increased CD8.4-Lck coupling. We respectfully disagree with the Reviewer that this argumentation is circular. This model enables us to address the overall outcome of the differentiation status + the higher reactivity to the (self)-antigens. In polyclonal VM vs. naïve T cells, both these two aspects are present as well. However, we modified this part of the manuscript in order to avoid any confusion.

The Reviewer is correct that the analysis of CD8 and CD8.4 cells should have been complemented with experiments that focus solely on the differentiation status. We decided to take advantage of our OVA-reactive clones (V14-C1, V14-C2) that naturally form both naïve and VM T cells at comparable ratios and established a second model for monoclonal naïve and VM T cells expressing the identical TCR. We compared these two cell types in the experimental model of autoimmune diabetes and observed that VM T cells were less efficient in the potential to induce the disease than their naïve counterparts with the same TCR (Fig. 6A).

In addition, is this really an autoimmune model? The OT-I T cells did not develop in a host that contained OVA or Q4H7. What they are really testing is whether Lm-OVA or Lm-L4H7 can prime transferred OT-I or OT-I;CD8.4 cells to enter the islets and kill OVA-expressing β -islet cells. For these T cells, OVA is a foreign antigen, not the low-affinity ligand driving positive selection and HSP (and presumably VM cell formation). The positive selecting ligand is known in the OT-I system (Hogquist KA. 1997. *Immunity*. 6:389) and examination of reactivity by naïve and VM cells against this ligand might be more informative. In the model used in this investigation, OVA simply represents a cognate foreign antigen to the transferred T cells, regardless of naïve or VM status.

The general problem of animal models for autoimmunity is that the real autoimmunity in humans develops for a long time, with relatively low incidence, is triggered largely by unknown genetic and environmental factors (with the exception of rare monogenic autoimmune disorders), and overall it is an incidental pathology/malfunction in the immune system. On the other hand, any animal model of autoimmunity must be reproducible, reliable, fast, and well controlled. For these reasons, each mouse model of autoimmunity has some issues, because it requires that at least one mechanism of self-

tolerance must be broken or bypassed in the experiment. In our case, it we bypassed central tolerance and clonal deletion. Along this line, OT-I cells (a clone that normally developed in a healthy WT mouse) does not get strongly activated by its positive selecting self-antigens (Salmond RJ et al. *Nat Immunol.* 2014 Sep;15(9):875-883; Oberle SG et al. *Cell Rep.* 2016 Oct 11;17(3):627-635.). If this was not the case, it would imply that WT C57Bl/6 mice would develop autoimmunity. We cannot exclude the possibility that some rare mice eventually develop autoimmunity caused by OT-I or similar T cell clones, but it is almost impossible to address this experimentally.

For priming of OT-I T cells, we mostly used Q4H7 as an antigen that does not induce negative selection of OT-I T cells (Daniels MA et al. *Nature.* 2006 Dec 7;444(7120):724-9.; Koehli S et al. *Proc Natl Acad Sci U S A.* 2014 Dec 2;111(48):17248-53.) and thus can potentially mimic a relatively strong, but still positively selecting antigen. Without the usage of this double transgenic system, we would not have been able to assess the level of tolerance of VM T cells on a per cell basis. Using this widely used experimental setup for T cell priming, we could assess the potential of naïve and VM T cells to induce autoimmune tissue pathology – i.e. the aggregate ability of these cells to expand, differentiate to effector T cells, infiltrate healthy tissues of the hosts, and cause the tissue damage.

For sure, these results cannot indicate whether any real-life autoimmune disease is induced by naïve or VM T cells and this was definitely not an ambition of our study. However, the CD8.4 VM (10 k transferred) were not able to induce the autoimmune pathology even when they were primed with their low-affinity cognate antigen Q4H7 in a mouse expressing OVA in the pancreas. I am convinced that this is a relatively solid evidence of their high level of self-tolerance, which is the conclusion of our *in vivo* experiments.

The Reviewer has a good point that it is worth testing the response of CD8.4/CD8WT OT-I T cells to the endogenous antigens. It has been reported that hyperreactive OT-I T cells lacking a phosphatase, *PTPN22*, but not WT OT-I T cells, do respond to the endogenous positive-selecting self-antigens (Salmond RJ et al. *Nat Immunol.* 2014 Sep;15(9):875-883.). Because the enhanced coreceptor-Lck coupling increases the reactivity of CD8.4 OT-I T cells, it might increase the responses to the endogenous antigens as well. We stimulated CD8WT OT-I and CD8.4 OT-I T cell by DCs loaded with the endogenous peptides *Catnb* and *Mapk8* *in vitro* and we did not observe any significant response in either of these clones (Fig. EV5E). We also analysed the response of the cells to Lm-*Catnb* and we did not see any difference when we compared it to empty Lm control infection (Fig. EV5F). These results go along with our conclusion that VM T cells have additional mechanisms that compensates their intrinsically high reactivity to self-antigens.

Although this manuscript describes interesting and important experiments with respect to virtual memory cells, I do not believe that it addresses the role of these cells in autoimmune processes.

This Reviewer is right that we did not address the role of VM T cells in real spontaneous autoimmune processes and we do not claim it in the manuscript. We investigated the level of self-tolerance of these cells by examining the potential of VM T cells to induce the pathology in our experimental model of autoimmune diabetes on a per cell basis. We changed some parts of the Abstract, Results, and Discussion of the revised manuscript to be more specific about this.

Referee #3:

The authors describe factors that determine the generation of virtual memory T cells and conclude that despite their self-reactivity and partial differentiation they do not seem more prone to trigger autoimmunity than naïve T cells. This study addresses the detailed characterization of memory-phenotype CD8 T cells and comes as a significant contribution to the body of literature accumulated since at least 12 years. What is really interesting about this manuscript is the high granularity of the molecular and functional characterization of virtual memory (VM) in comparison to naïve and true memory T cells, using cutting edge tools and animal models. As the authors point out, the novelty of the present study stems from their showing how the T cell intrinsic sensitivity to TCR-originated signals determines the magnitude of naïve T cell transit to virtual memory T cells. Moreover, the RNA-seq analyses convincingly place the VM subset as an intermediate between naïve and central memory T cells. In short this report may significantly enrich the description of the phenotypical makeup and functional competence of memory-phenotype CD8 T cells.

This study is well performed and clearly written. A minor concern is the repetitious nature of the discussion section relative to the content of the results. It may strengthen the manuscript to avoid too much repetition in the discussion. In addition, the authors may discuss how these high resolution-defined VM T cells relate to the so called stem like memory T cells. The recent reports on both mouse and human stem cell like memory T cells allow to place this subset between naïve and memory T cells. Some of the phenotypic traits for the latter cells are shared with those of VM T cells. An so does the overall functional competence they have displayed. It would be interesting for the readers following the characterization of T cell memory to know how the authors may bridge this "new" subset with the VM T cell subset. There is a need for integration and streamlining of the memory cell subsets.

We are very thankful to the Reviewer for the positive assessment of our work. We absolutely agree with the Reviewer that the classification of various T-cell subsets should be revised and streamlined. The Reviewer is correct that VM T cells and stem-like memory (SCM) T cells exhibit some common traits including higher expression of CD122, CXCR3, and dependency on IL-15 (Zhang Y et al. Nat Med. 2005 Dec;11(12):1299-305.). However, unlike VM cells, SCM T cells are derived from CD44^{low} population which also applies for human SCM T cells counterpart derived from CD45RA⁺ population (Gattinoni L et al. Nat.Med. 2011 Sep; 17(10):1090-97.). Furthermore, SCM T cells express Sca-1 stemness marker which is not upregulated in the VM T cells as revealed by our RNAseq data. Based on self-renewing potential attributed to SCM T cells and the capacity to give rise to central memory, effector memory and effector CD8⁺ T cells, we cannot exclude that VM T cell population preferentially arise from SCM T cells. However, VM and SCM T cells represent two distinguishable subsets of CD8⁺ T cells. We have modified the discussion section appropriately.

Minor points:

Page 3, paragraph 3, numeral (ii) is duplicated.

We are thankful to this Reviewer for finding this typo. We fixed it.

Figure 3G: the nomenclature of the naïve TCR retrogenics differs slightly from that provided in the text (page 7, next to last paragraph).

We have corrected this issue.

Thank you for submitting your revised manuscript to the EMBO Journal. Your revision has now been seen by the original referees and their comments are provided below.

As you can see the referees appreciate the revisions and support publication. I am therefore very please to accept the manuscript for publication in The EMBO Journal.

REFERE REPORTS

Referee #1:

The authors responded well to this reviewer's previous concerns, providing new data and revisions to the manuscript that strengthen the main conclusions and addressed all major issues.

As a recommendation - in response to points raised by other reviewers - the inherent difficulty in dissociating the effects of CD8.4 on both VM phenotype and hyper-reactivity may potentially be addressed in FUTURE work by breeding TCR transgenic and CD8.4 alleles onto the IL-15^{-/-} background. Based on previous studies, one might expect that this will prevent the establishment and/or survival of VM cells, leaving behind naive cells with the hyper-reactivity brought by the CD8.4 allele, allowing this variable to be investigated in isolation. This is similar to the approach the authors used for Fig. 6 (although in that case, it is unclear what is the basis by which some cells have remained of naive phenotype while most others have attained VM phenotype, complicating interpretation).

Again, this comment is provided as unsolicited advice, not with the suggestion that such time-consuming experiments are needed for the current manuscript, which already covers considerable ground.

Referee #2:

The authors have responded thoughtfully and extensively to my concerns and those of the other reviewers. I do not have any further concerns.

Referee #3:

The authors have carefully addressed most of the points and concerns raised by the three reviewers, mine included. I believe that the revised manuscript is suitable for publication in The EMBO Journal.

Corresponding Author Name: Ondrej Stepanek

Manuscript Number: EMBOJ-2017-98518R